# A synaptic mechanism for encoding the learned value of action-derived safety

Emma E. Macdonald[1,6] ✉, Jun Ma[1,5,6], Di Liu[1,6], Kai Yu[1,6], Rachel A. Walker[1], Michael E. Authement[2], Yan Leng[1], Hannah C. Goldbach [2], Guochuan Li [3], Yulong Li [3], Veronica A. Alvarez [2], Bruno B. Averbeck [4] ✉ & Mario A. Penzo [1] ✉

Adaptive behavior requires that behaviorally relevant signals gain access to neural circuits guiding action. The thalamus has long been proposed to regulate information flow to cortical and subcortical systems, yet whether it also tracks internally generated goal signals remains unclear. Here, we show that neurons in the paraventricular thalamus (PVT) projecting to the nucleus accumbens (NAc) encode the motivational value of safety during active avoidance. As mice learn to avoid threat, PVT→NAc neurons develop a signal that emerges selectively at successful avoidance, is experience-dependent, and diminishes following outcome devaluation. Selective silencing of the PVT→NAc pathway at safety onset reduces the motivational value assigned to safety without impairing action–outcome learning. Mechanistically, PVT input engages cholinergic interneurons (CIN) in the NAc to regulate dopamine release via synaptic potentiation mediated by GluA2-lacking AMPA receptors at PVT–CIN synapses. Disrupting this plasticity reduces the motivational impact of safety. These findings identify a thalamostriatal mechanism through which learned goals gain stable access to motivational circuitry.

Adaptive behavior depends on the ability to represent and update goals and values that guide future actions[1-3]. A central challenge in neuroscience has been to understand how the brain tracks what is currently important for behavior, particularly when goals must be inferred from experience, internal state, or recent outcomes rather than specified by immediate sensory cues. While much of this work has focused on cortical and striatal mechanisms underlying valuation and decision-making[1,3-5], influential theoretical frameworks have long suggested that the thalamus may play a broader role in shaping how goal-relevant information is represented and utilized across distributed brain networks[6-8]. How this thalamic contribution extends beyond sensory processing to the tracking of motivationally relevant goals, however, remains poorly understood.

Safety is an internal state derived from the successful prevention of harm, and a prominent example of an internally valued goal that must be derived from context, past actions, and learned cues[9-11]. Importantly, safety can be generated instrumentally: in active avoidance, the absence of an aversive outcome becomes meaningful only because it is contingent on the animal's action[12-14]. Instrumentally generated safety differs fundamentally from passive safety cues that merely signal threat omission; instead, it promotes behavioral persistence, reinforces future avoidance, and engages dopaminergic systems in a manner analogous to reward[10,15,16]. Despite its central role in adaptive behavior, how action-derived safety is tracked and assigned motivational value within neural circuits remains incompletely understood.

[1]Section on the Neural Circuits of Emotion and Motivation, National Institute of Mental Health, Bethesda, MD, USA. [2]Section on Neurobiology of Compulsive Behaviors, National Institute of Mental Health, Bethesda, MD, USA. [3]State Key Laboratory of Membrane Biology, School of Life Sciences, Peking University, Beijing, China. [4]Laboratory of Neuropsychology, National Institute of Mental Health, Bethesda, MD, USA. [5]Present address: Jiangsu Province Key Laboratory of Anesthesiology, Xuzhou Medical University, Xuzhou, China. [6]These authors contributed equally: Emma E. Macdonald, Jun Ma, Di Liu, Kai Yu. ✉e-mail: emma.macdonald@nih.gov; averbeckbb@mail.nih.gov; mario.penzo@nih.gov

Midline thalamic nuclei are anatomically positioned to contribute to this problem by linking internal state signals with limbic and striatal circuits that regulate motivation and action[17-20]. Among these, the paraventricular thalamus (PVT) has emerged as a prominent node, receiving convergent input from brainstem and hypothalamic regions conveying visceral and homeostatic information, as well as from cortical areas involved in contextual and cognitive processing[18,21]. This convergence positions the PVT to link internal state signals with information about recent actions and outcomes—a key requirement for tracking the motivational significance of learned goals[22]. Consistent with this organization, prior work has implicated PVT circuits in drug seeking and relapse, modulation of reward-related behavior, and defensive control[21,23-27]. However, whether PVT projections participate directly in tracking goal signals that arise through learning, such as action-derived safety, remains unclear.

Here, we leveraged an active avoidance paradigm to examine how the PVT contributes to the motivational representation of safety. Focusing on projections from the PVT to the nucleus accumbens (NAc), a key node in motivational control[28-30], we combined pathway-specific optogenetic perturbations, fiber photometry, synaptic physiology, and computational modeling. We show that PVT→NAc projections develop an experience-dependent signal linked to successful avoidance, which is necessary for maintaining the motivational impact of safety but not for acquiring instrumental actions. Mechanistically, this process depends on plastic interactions between PVT inputs and striatal cholinergic interneurons that regulate dopaminergic tone. Together, these findings identify a thalamostriatal mechanism by which the midline thalamus tracks action-derived safety as a motivationally relevant goal, providing insight into how internal goals acquire value and guide adaptive behavior.

## Results

### PVT → NAc projections encode the learned value of action-derived safety

To determine how safety outcomes acquire motivational significance during avoidance learning, we trained mice in a two-way active avoidance (2AA) task in which they learned to shuttle across a hurdle during a warning signal (WS) to terminate the WS and avoid a predicted footshock (see Methods). Classical and contemporary accounts of avoidance learning have emphasized that avoidance actions generate response-contingent stimuli, such as WS offset, that are negatively correlated with shock and can serve as reinforcing feedback for avoidance behavior[31]. While such response-generated sensory events are an important feature of avoidance tasks, behavioral performance in 2AA depends strongly on action–outcome reliability and predictability even when these sensory consequences are held constant (Supplementary Fig. 1). This indicates that behavioral performance reflects the learned motivational significance of successful outcomes rather than the presence of any single sensory event per se.

To examine how this motivational signal emerges at the circuit level, we recorded calcium dynamics from GCaMP8s-expressing terminals of PVT neurons in the NAc during training (days 1, 3, and 5; Fig. 1A, B; Supplementary Fig. 2A). Population heatmaps aligned to WS onset and sorted by escape latency revealed activity patterns that spanned the cue period and increased with learning (Fig. 1C, D). However, inspection of individual trials suggested more discrete increases in PVT→NAc activity time-locked to specific behavioral events, including shuttle initiation and the moment associated with successful avoidance (Fig. 1E), as previously reported[27]. Aligning activity to these events confirmed distinct peaks at both time points (Fig. 1F,G). We therefore focused subsequent analyses on the avoidance-related signal emerging at successful outcomes, consistent with the engagement of PVT→NAc projections during the motivational evaluation of having avoided harm.

To test the functional relevance of this outcome-related signal, we used closed-loop optogenetic silencing of PVT→NAc terminals immediately following successful avoidance. Across training days, inhibition of this pathway impaired the development of stable avoidance performance relative to controls (Fig. 1H–J, Supplementary Fig. 2B, C). Notably, this deficit persisted during a subsequent light-off probe session in which no optogenetic manipulation was delivered, indicating that the behavioral effect reflects a lasting alteration rather than a transient disruption of performance (Fig. 1J). To probe the underlying computational mechanism, and to dissociate learning of action-outcome associations from the assignment of motivational value to successful avoidance, we applied a reinforcement learning model in which action learning and outcome value learning were estimated separately (Fig. 1K, see Methods). This analysis revealed that behavioral deficits following PVT→NAc inhibition were best explained by a selective reduction in the value of the safety outcome, rather than a change in the action learning rate (Fig. 1L). A complementary Bayesian learning model similarly revealed subject-level decreases in estimated safety value (Supplementary Fig. 3). Conversely, optogenetic excitation of PVT→NAc terminals during the safety period increased modeled outcome value while suppressing action learning rate (Supplementary Fig. 4), suggesting that excessive activation produces broader effects than temporally precise inactivation. Nevertheless, fitting the model at the individual subjects revealed a consistent trend toward increased valuation (Supplementary Fig. 4F, G), reinforcing the conclusion that this projection primarily encodes outcome value rather than directly instructing action selection.

We next asked whether this safety-related signal was sensitive to devaluation of instrumental contingency, a key criterion for distinguishing value encoding from fixed stimulus–response representations. Trained animals underwent an extinction-with-response-prevention session in which a transparent barrier blocked shuttling, and the WS no longer predicted shock. When subsequently tested in extinction sessions (with the barrier removed and no shock delivered), mice exhibited an approximately 50% reduction in avoidance behavior. This behavioral change was accompanied by a significant decrease in the safety-evoked PVT→NAc signal (Supplementary Fig. 5A–D). When the shock contingency was reinstated, both avoidance behavior and PVT→NAc safety-related activity recovered, often surpassing pre-devaluation levels (Supplementary Fig. 5E). This recovery further supports the interpretation that extinction-related effects arose from the contingency-dependent value updating and not attributable to photobleaching or signal degradation.

Finally, we tested whether this thalamic-derived value signal supports effortful avoidance behavior. Parametrizing effort in shuttle-based avoidance is challenging, as animals either cross or fail to cross without a clear gradient of work required. To overcome this limitation, we employed a head-fixed version of the task in which mice learned to run a fixed distance to avoid an air puff (Fig. 1M). In this setting, within-subject silencing of PVT→NAc terminals reduced the break point in a progressive ratio test (see Methods), indicating a diminished willingness to work for safety (Fig. 1N, O). Together, these results demonstrate that PVT→NAc projections encode a safety-related value signal that emerges with learning and sustains motivated behavior by supporting the motivational value of safety under increasing effort demands.

### PVT → NAc projections are required for safety-evoked dopamine release

Previous work has shown that dopamine (DA) levels in the NAc increase in response to safety-associated events during avoidance behavior, consistent with the idea that threat omission can acquire reward-like motivational value[15,32-35]. We first asked whether a similar dopaminergic signal accompanies successful avoidance in our 2AA task, and whether its emergence depends on input from the PVT. To

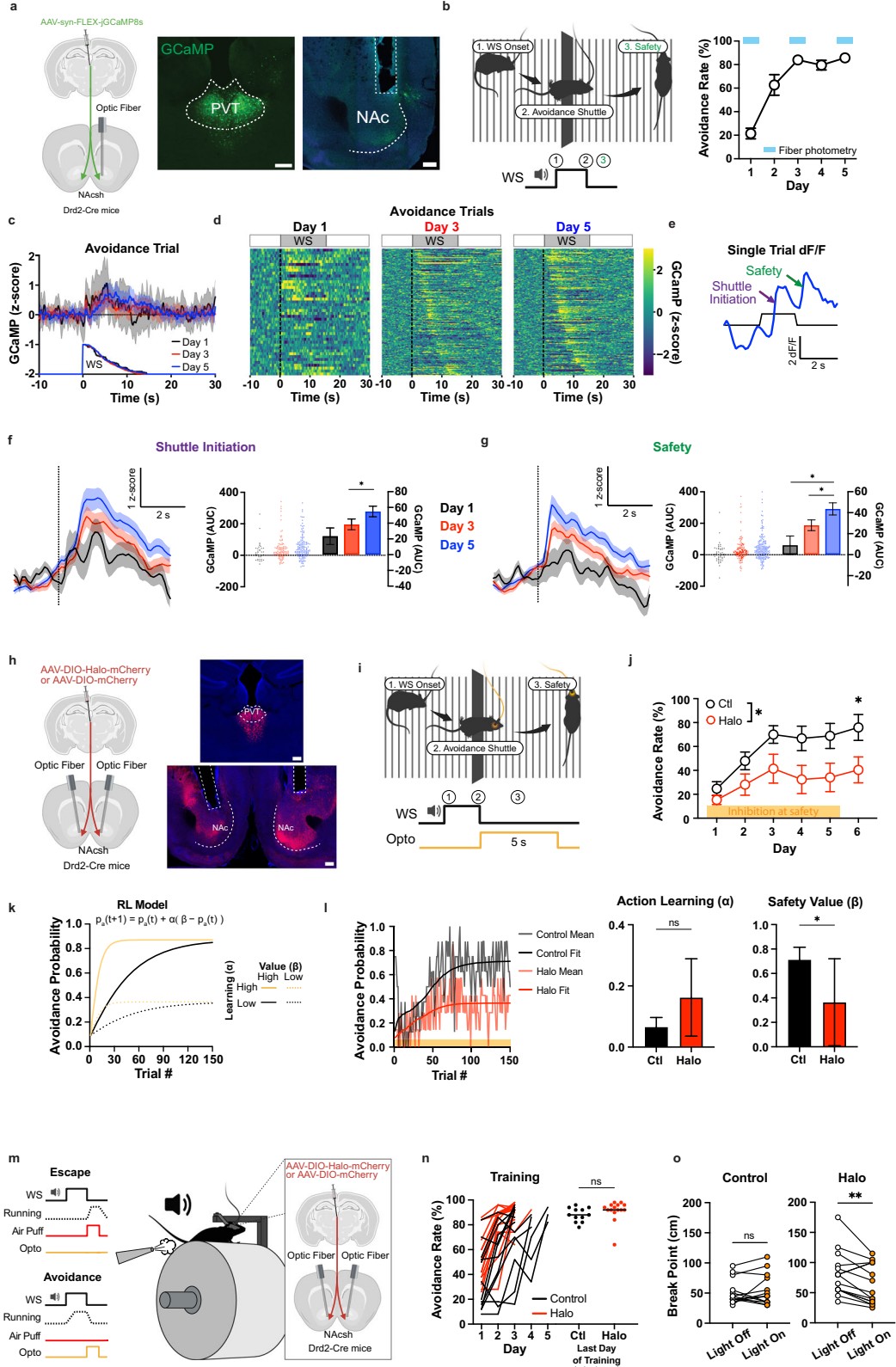

address this, we expressed the genetically encoded DA sensor dLight1.2[36] in the NAc and recorded DA dynamics during training. As animals acquired stable avoidance performance, we observed a progressive increase in DA release time-locked to the onset of the safety period following successful avoidance (Fig. 2A–F; Supplementary Fig. 6). Although the across-day effect in the dLight cohort did not reach conventional statistical significance in an omnibus mixed-effects

test, the effect size and direction were consistent with a training-dependent strengthening of the safety-associated DA response. Importantly, this signal emerged gradually with learning, arguing against a fixed sensory response and instead suggesting the acquisition of motivational value.

We next tested whether this safety-related dopaminergic signal depends on thalamic input. During dLight recordings, we performed

**Fig. 1 | PVT → NAc projections encode the value of inferred safety and sustain motivation during avoidance. a** Injection schematic (left). Representative histology (right). Scale bars, 200 μm. **b** Schematic of 2AA avoidance trial (left). Performance in the 2AA task ($n = 7$ mice) (right). **c** GCaMP dF/F z-score normalized in all avoidance trials by day. **d** Heatmaps of calcium signals during avoidance trials, ordered by avoidance latency (low to high) for Days 1 (44 trials), 3 (176 trials), and 5 (180 trials). **e** Representative dF/F signal during avoidance trial. **f,g** GCaMP dF/F z-score normalized at shuttle initiation (**f**) (dotted line) or safety (**g**) in avoidance trials (left). Quantification of post-event activity (right) with individual trials on the left y-axis and mean±s.e.m. on the right y-axis. AUC, pairwise comparison across days, mixed effects model (see statistics in Supp. Data 1): Shuttle initiation: $n = 288$ trials from 7 mice; Safety: $n = 401$ trials from 7 mice. Asterisks indicate where multiple comparisons (Sidak corrected) found significance. **h** Injection schematic (left). Representative histology (right). Scale bars, 200 μm. **i** Schematic of closed-loop optogenetic manipulation (see Methods). **j** Performance in the 2AA task (Control $n = 8$, Halo $n = 7$ mice). 2-way Anova group effect days 1-5: $p = 0.0345$, Mann-Whitney test day 6: $p = 0.0135$. **k** Simulated avoidance probability from RL model ($p_a$=probability of avoidance, β=value of safety, α=learning rate). **l** Avoidance probability across trials with best-fit model probability by group (left). Best-fit learning (middle) and value parameter (right) by group with bootstrap-estimated s.e.m. Bootstrap test for group difference: Learning: $p = 0.1032$; Value: $p = 0.0448$ (Control $n = 8$, Halo $n = 7$ mice). **m** Schematic of head-fixed avoidance and injections (see Methods). **n** Performance in avoidance training with no stimulation (left) and last day of training (right, Unpaired t test: $p = 0.2714$, Control $n = 10$, Halo $n = 13$ mice). **o** Breakpoints for progressive ratio tests for Halo (left, Paired t test: $p = 0.0079$, $n = 13$) and mCherry (right, Paired t test: $p = 0.4947$, $n = 10$) groups. Data are shown as mean ± s.e.m. *$p < 0.05$, **$p < 0.01$, ns: $p > 0.05$. Schematics created in BioRender. Macdonald, E. (2026) https://BioRender.com/4pm04zs. Source data are provided as a Source Data file. All statistical tests are two-sided.

unilateral closed-loop optogenetic silencing of PVT→NAc terminals selectively at the time of successful avoidance (Fig. 2G). This manipulation did not alter avoidance performance (Fig. 2I), indicating that changes in DA signaling were not secondary to behavioral disruption. Nevertheless, silencing PVT→NAc projections significantly attenuated the avoidance-related DA response in the NAc (Fig. 2J,K). Together, these results indicate that the emergence of a dopaminergic signal linked to successful avoidance depends on PVT input. Rather than reflecting a purely sensory correlate of task structure, safety-related DA release appears to be gated by thalamostriatal signaling, consistent with a role for the PVT in coupling learned, action-dependent outcomes to motivational circuitry in the NAc.

## Cholinergic interneurons mediate PVT-driven dopamine signaling

Although PVT projections are required for safety-related DA release in the NAc, prior anatomical studies suggest that midline thalamic inputs, including those from PVT, do not directly innervate dopaminergic terminals in the striatum[37]. Instead, these projections densely innervate striatal cholinergic interneurons (CINs), which are well positioned to regulate DA release through nicotinic receptor-dependent mechanisms[38–45]. Consistent with this organization, recent circuit-mapping work identified that the PVT is a major source of excitatory input to NAc CINs[39], suggesting a plausible substrate for PVT-dependent modulation of dopaminergic signaling.

To test whether PVT input can evoke DA release through cholinergic mechanisms, we performed fast-scan cyclic voltammetry (FSCV) in acute NAc slices while optogenetically stimulating PVT terminals. PVT stimulation reliably evoked DA release, which was abolished by bath application of the nicotinic antagonist DHβE (Fig. 3A-C). These results indicate that PVT-evoked DA release requires intact cholinergic transmission.

We next asked whether CINs are themselves engaged during the acquisition of safety value in vivo. Using the genetically encoded ACh sensor gACh3.8[46], we recorded cholinergic activity in the NAc across training in the 2AA task. Successful avoidance was associated with a phasic ACh response that increased in magnitude with training (Fig. 3D-J; Supplementary Fig. 6), closely paralleling the emergence of safety-related signals observed in PVT terminals and DA release. These data suggest that CIN recruitment during safety emerges with learning and may participate in value assignment rather than reflecting a fixed sensory response.

To directly test whether CIN activity is required for PVT-evoked DA release in vivo, we expressed the DA sensor dLight1.2 in the NAc of ChAT-Cre mice, optogenetically stimulated PVT terminals, and simultaneously silenced CINs using the *D*esigner *R*eceptors *E*xclusively *A*ctivated by *D*esigner *D*rugs (DREADDs) hM4Di. PVT-evoked DA was markedly potentiated by avoidance training, indicating that learning enhances the efficacy of this circuit. Strikingly,

chemogenetic inhibition of CINs completely abolished PVT-evoked DA release following training (Fig. 3K-O), providing convergent evidence that CINs are necessary for translating PVT input into dopaminergic output.

Finally, to examine the temporal relationship between cholinergic and dopaminergic signaling during safety, we performed simultaneous recordings using spectrally distinct fluorescent sensors. Because prior experiments relied on the green DA sensor dLight1.2, we first validated the specificity and temporal fidelity of a red-shifted GRAB-DA sensor (rDA1m)[47] to enable dual-color recordings. We expressed rDA1m in the NAc and simultaneously recorded calcium activity from GCaMP8s-expressing dopaminergic terminals originating in the ventral tegmental area (VTA) (Supplementary Fig. 7A-J). Across animals and behavioral epochs, rDA1m signals closely tracked VTA dopaminergic terminal activity, exhibiting strong trial-by-trial correspondence and significant Pearson correlations, consistent with rDA1m reliably reporting local DA dynamics in the NAc.

Having established the validity of rDA1m, we next co-expressed rDA1m and the acetylcholine sensor gACh3.8 in the NAc to simultaneously monitor dopaminergic and cholinergic signaling during avoidance behavior. Both sensors recapitulated the safety-related transient observed in single-sensor experiments (Supplementary Fig. 7K-U). In this independent cohort, safety-associated DA responses increased significantly across training days, consistent with the progressive modulation observed in the original dLight recordings. Notably, within-trial analyses revealed a consistent temporal offset during the safety period, such that cholinergic transients preceded dopaminergic peaks. This temporal relationship became more pronounced with training, paralleling the emergence of safety-related value signals in PVT inputs and DA release (Supplementary Fig. 7K–U).

Together, these results indicate that safety learning enhances a sequential signaling cascade in which PVT-driven engagement of striatal cholinergic interneurons precedes and facilitates dopaminergic release in the NAc.

## Safety value learning induces synaptic plasticity at PVT → CIN synapses

The potentiation of PVT-evoked, CIN-dependent DA release in the NAc suggests that avoidance learning engages experience-dependent plasticity within the PVT → CIN microcircuit. We therefore asked whether the enhanced functional influence of PVT inputs following training reflects synaptic modifications at PVT → CIN connections. To test this, we injected ChAT-Cre mice with a Cre-dependent mCherry reporter in the NAc to label CINs and expressed ChR2-eYFP in PVT neurons to enable selective optical activation of PVT→NAc terminals (Fig. 4A). Mice were then trained in the 2AA task or assigned to a yoked control group that received identical cue and shock exposure but lacked control over shock omission (Fig. 4B). As expected, only trained

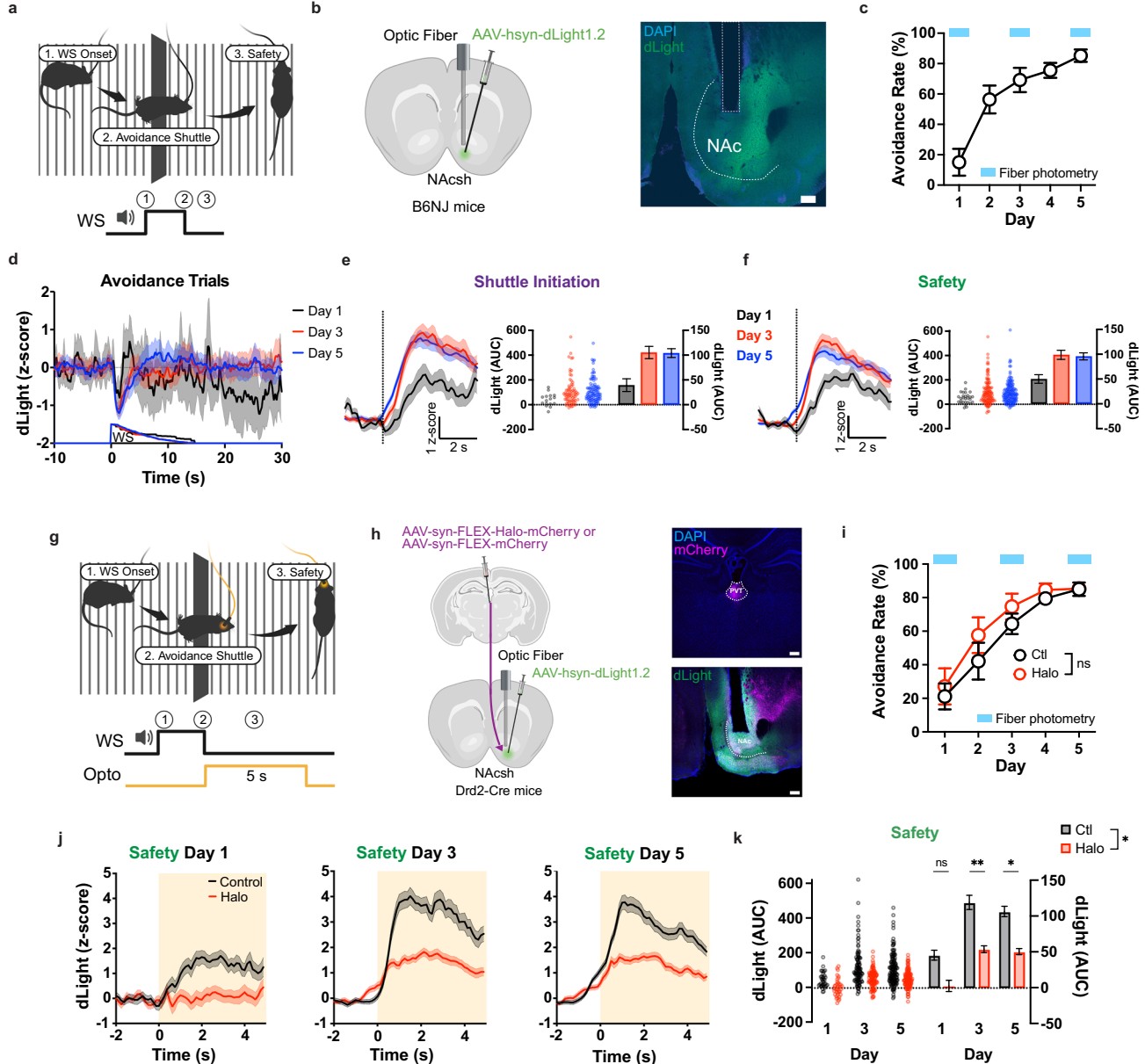

**Fig. 2 | PVT input drives NAc dopamine signals associated with safety.**
**(a)** Schematic of behavior apparatus and example avoidance trial depicting behavior sequence at WS onset. **(b)** Wild type mice were injected with dLight1.2 to record dopamine in NAc (left). Representative image of recording site (right). Scale bar, 200 µm. **(c)** Performance in the 2AA task (n = 6 mice). **(d)** dLight signal in all avoidance trials by day. **(e-f)** dLight dF/F z-score normalized at shuttle initiation (dotted line) **(e)** or safety **(f)** in avoidance trials (left). Quantification of post event activity (right) with individual trials on the left y-axis and mean ± s.e.m. on the right y-axis. AUC, pairwise comparison across days, mixed effects model (see statistics in Supp. Data 1): Shuttle initiation: n = 204 trials from 6 mice; Safety: n = 305 trials from 6 mice. **(g)** Schematic of closed-loop optogenetic manipulation in the 2AA task (see Methods). **(h)** Drd2-Cre mice were injected with halorhodopsin (Halo) or fluorescent control in PVT and dLight1.2 in NAc to test the effects of PVT-NAc

inhibition of safety dopamine release (left). Representative images of injection and recording sites (right). Scale bars, 200 µm. **(i)** Performance in the 2AA task. 2-way Anova: p = 0.517 (Control n = 6, Halo n = 5 mice). **(j)** dLight dF/F z-score normalized at safety with optogenetic inhibition by group day 1 (left), day 3 (middle), and day 5 (right) of 2AA training. **(k)** Quantification of post event activity with individual trials on the left y-axis and mean ± s.e.m. on the right y-axis. AUC, pairwise comparisons between groups, linear mixed-effects model for repeated measures: (see statistics in Supp. Data 1): Control n = 303 trials from 6 mice; Halo n = 277 trials from 5 mice. Asterisks indicate where multiple comparisons (Sidak corrected) found significance. Data are shown as mean ± s.e.m. *p < 0.05, **p < 0.01, ns: p > 0.05. Schematics created in BioRender. Macdonald, E. (2026) https://BioRender.com/4pm04zs. Source data are provided as a Source Data file. All statistical tests are two-sided.

animals developed robust avoidance behavior by day 5, whereas yoked controls failed to acquire shuttle responses to the WS (Fig. 4C). Two hours after the final training session (Day 5), acute coronal brain slices containing the NAc were prepared, and optogenetically evoked excitatory postsynaptic currents (oEPSCs) were recorded from fluorescently identified CINs. Compared to yoked controls, CINs from trained animals exhibited significantly larger AMPA receptor-mediated oEPSCs (Fig. 4D,E) and an elevated AMPA/NMDA ratio (Fig. 4I), consistent with

synaptic potentiation. In contrast, NMDA receptor-mediated current amplitudes were unchanged between groups (Fig. 4H), indicating that plasticity was expressed postsynaptically, as a selective increase in AMPA receptor function.

We next examined the locus and mechanism of this synaptic enhancement. CINs from trained animals showed significantly faster AMPA current rise and decay kinetics relative to controls (Fig. 4F,G), consistent with the recruitment of GluA2-lacking, calcium-permeable

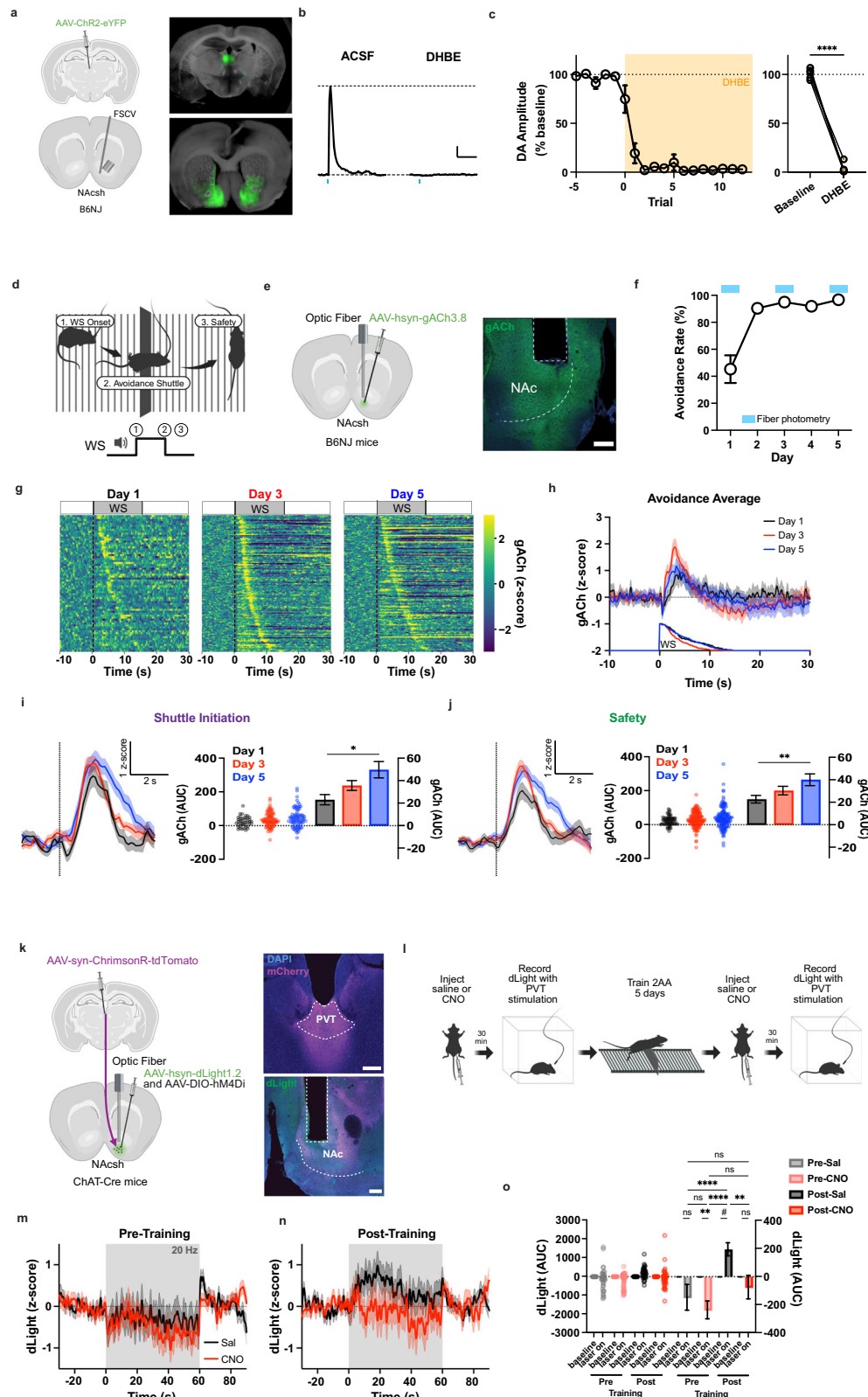

AMPA receptors[48–50]. In line with this interpretation, rectification indices were increased in CINs from trained mice, and AMPA currents were more strongly attenuated by NASPM, a selective blocker of GluA2-lacking AMPARs (Fig. 4K,L). In addition, a modest but significant reduction in paired-pulse ratio was observed following training (Fig. 4J), suggesting that presynaptic changes in release probability may also contribute to the overall potentiation.

Together, these results demonstrate that active avoidance training induces synaptic strengthening at PVT → CIN synapses in the NAc, characterized by enhanced AMPA receptor transmission and recruitment of GluA2-lacking AMPARs. This synaptic plasticity provides a mechanistic substrate through which learning enhances the ability of PVT inputs to recruit CINs and, in turn, facilitate safety-related dopamine signaling.

**Fig. 3 | NAc acetylcholine is necessary for PVT-driven dopamine signaling after safety learning. (a)** Wildtype mice injected with channelrhodopsin (ChR2) in PVT and fast-scan cyclic voltammetry (FSCV) performed in NAc slices (right). Representative images of injection and recording sites with ChR-eYFP expression (right). **(b)** Representative PVT-evoked [DA] in ACSF and DHBE. Scale, 100 nM, 2 s. **(c)** oDA amplitude normalized to baseline in ACSF and DHBE (left). Average oDA amplitude of three trials by slice (right). Paired t test: p < 0.0001 (n = 5 slices from 5 mice). **(d)** Schematic of 2AA avoidance trial. **(e)** Wildtype mice were injected with GRAB-ACh3.8 (gACh) in NAc (left). Representative image of recording site (left). Scale bar, 200 μm. **(f)** Performance in the 2AA task (n = 5 mice). **(g)** Heatmaps of gACh signals during avoidance trials, ordered by avoidance latency (low to high) for Days 1 (70 trials), 3 (143 trials), and 5 (145 trials). **(h)** gACh signal in all avoidance trials by day. **(i-j)** gACh dF/F z-score normalized at shuttle initiation (dotted line) **(i)** or safety **(j)** in avoidance trials (left). Quantification of post-event activity (right) with individual trials on the left y-axis and mean±s.e.m. on the right y-axis. AUC, pairwise

comparison across days, mixed effects model (see statistics in Supp. Data 1): Shuttle initiation: n = 220 trials from 5 mice; Safety: n = 348 trials from 5 mice. Asterisks indicate where multiple comparisons (Sidak corrected) found significance. **(k)** ChAT-Cre mice injected with ChrimsonR in PVT and dLight1.2 and DIO-hM4Di in NAc (left). Representative images of injection and recording sites (right). Scale bar, 200 μm. **(l)** Schematic of recording and training schedule (see Methods). **(m-n)** dLight during PVT stimulation pre-training **(m)** and post-training **(n)**. **(o)** Quantification of dLight activity with individual trials on the left y-axis and mean ±s.e.m. on the right y-axis. AUC, mixed effects model (see statistics in Supp. Data 1): n = 160 trials from 4 mice. Asterisks indicate where multiple comparisons (Sidak corrected) found significance. Data are shown as mean±s.e.m. *p < 0.05, **p < 0.01, ****p < 0.0001, ns: p > 0.05, #: p < 0.08. Schematics created in BioRender. Macdonald, E. (2026) https://BioRender.com/4pm04zs. Source data are provided as a Source Data file. All statistical tests are two-sided.

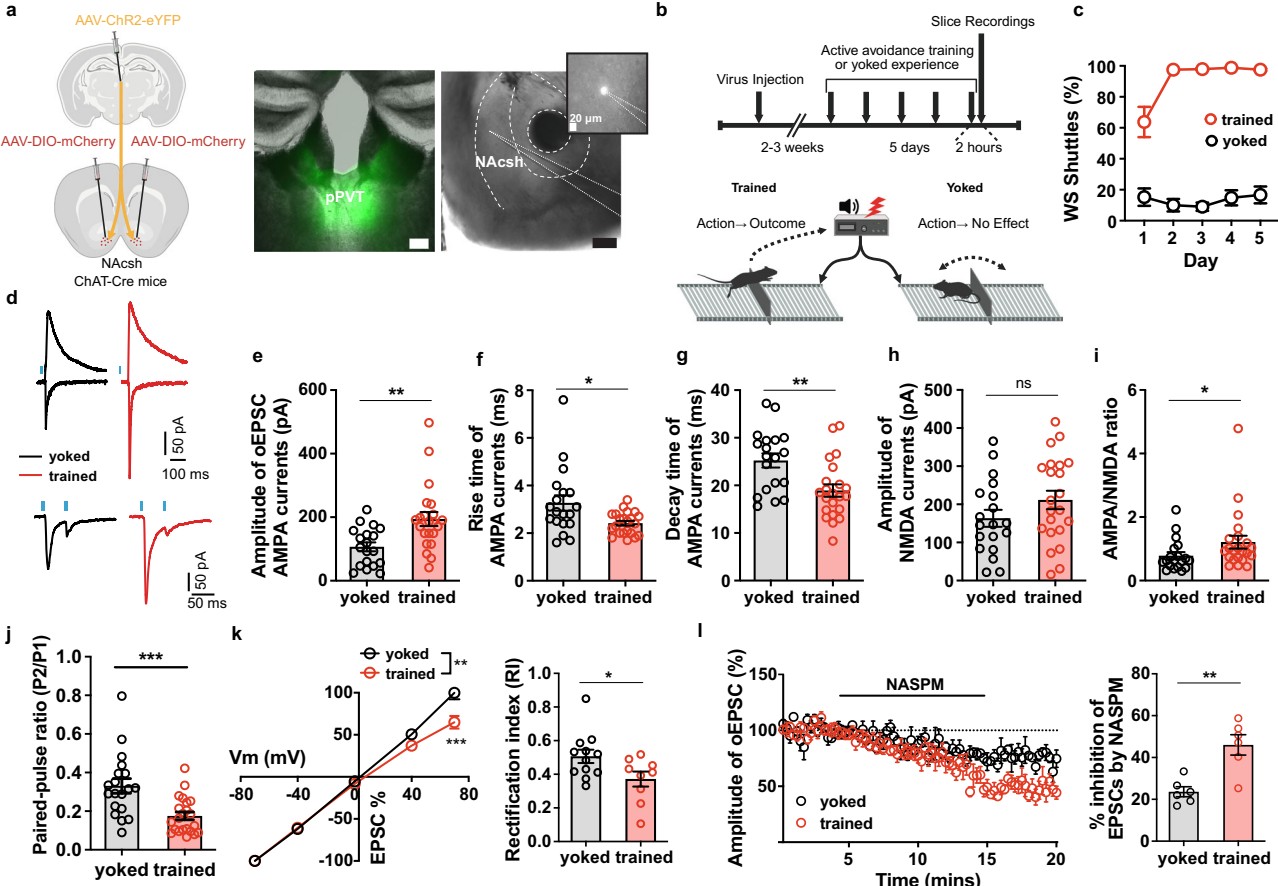

**Fig. 4 | Active avoidance training induces plasticity at PVT → CIN synapses. (a)** ChAT-Cre mice were injected with channelrhodopsin in PVT and DIO-mCherry in NAc to record post-synaptic currents in CINs with PVT stimulation (left). Representative images of injection and recording sites (right). Scale bars, 200 μm. **(b)** Schematic of training and recording schedule (see Methods). **(c)** Performance in the 2AA task (trained n = 8, yoked n = 7 mice). **(d)** Representative traces of optogenetic evoked voltage clamp responses. **(e-j)** Measurements of current properties. Unpaired t tests: oEPSC amplitude p = 0.0024; Rise time p = 0.0107; Decay time p = 0.0028; NMDA amplitude p = 0.1562; Paired-pulse ratio p = 0.0005. Mann Whitney test: AMPA/NMDA ratio p = 0.0168 (trained n = 22 neurons from 4 mice,

yoked n = 19 neurons from 3 mice). **(k)** Rectification curve (left). 2-way Anova: p = 0.0085; Bonferroni's multiple comparisons at 70 mV: p < 0.0001. Rectification index (right). Unpaired t test: p = 0.0360. **(l)** Amplitude of evoked response with NASPM application (left). Quantification of NASPM inhibition (right). Unpaired t test: p = 0.002 (trained n = 6 neurons from 3 mice, yoked n = 6 neurons from 2 mice). Data are shown as mean ± s.e.m. *p < 0.05, **p < 0.01, ***p < 0.001, ns: p > 0.05. Schematics created in BioRender. Macdonald, E. (2026) https://BioRender.com/4pm04zs. Source data are provided as a Source Data file. All statistical tests are two-sided.

## GluA1-dependent plasticity at PVT → CIN synapses supports safety value encoding

The preceding experiments showed that avoidance training potentiates PVT → CIN synapses through the recruitment of GluA2-lacking AMPA receptors, a form of plasticity classically associated with the insertion of GluA1-containing AMPA receptor[51,52]. We therefore asked

whether GluA1 expression in CINs is required for this experience-dependent synaptic remodeling. To test this, we used a CRISPR-Cas9 strategy to selectively delete *Gria1*, the gene encoding GluA1, from cholinergic interneurons. ChAT-Cre mice were injected unilaterally with a Cre-dependent CRISPR construct (AAV9-EF1α-DIO-Cas9-sgGria1) that co-expresses Cas9 and a guide RNA targeting *Gria1*[53],

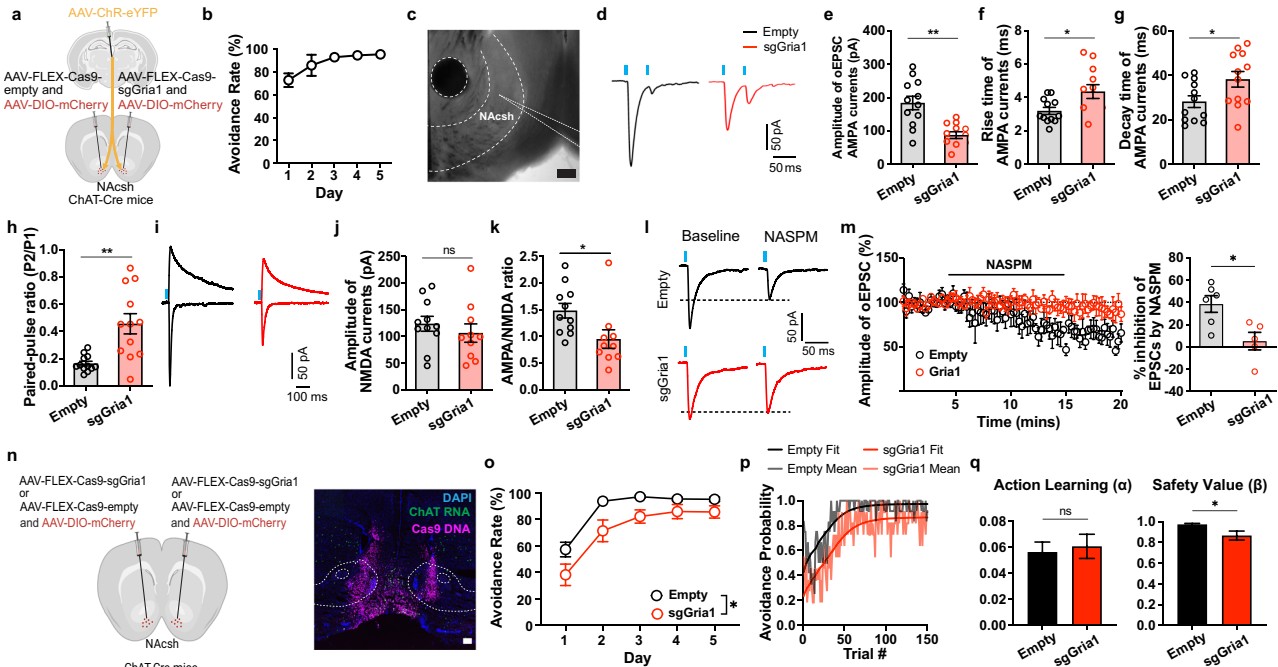

**Fig. 5 | NAc CIN Gria1 knockout disrupts avoidance-related plasticity and safety valuation. (a)** ChAT-Cre mice were injected with channelrhodopsin in pPVT, and FLEX-Cas9-sgGria1 and empty control contralaterally in NAc to record post-synaptic currents in CINs with pPVT stimulation (left). **(b)** Performance in the 2AA task (n = 5 mice). **(c)** Representative image of recording location. Scale bar, 200 μm. **(d)** Representative oEPSCs from control and knockout hemispheres. **(e-h)** Measurements of current properties. Unpaired t tests: oEPSC amplitude p = 0.0003; Rise time p = 0.0206; Decay time p = 0.0321; Paired-pulse ratio p = 0.0009 (sgGria1 n = 12 neurons, empty n = 12 neurons from 4 mice). **(i)** Representative traces of AMPA and NMDA responses to pPVT optogenetic stimulation. **(j-k)** Current amplitude measurements. Unpaired t tests: NMDA amplitude p = 0.4374; AMPA/NMDA ratio p = 0.0256 (sgGria1 n = 12 neurons, empty n = 12 neurons from 4 mice). **(l)** Representative traces of voltage clamp responses to optogenetic stimulation before and after NASPM application. **(m)** Amplitude of evoked response with NASPM application (left). Quantification of NASPM inhibition (right). Unpaired t test: p = 0.0135 (sgGria1 n = 5 neurons, empty n = 6 neurons from 5 mice). **(n)** ChAT-Cre mice were injected with FLEX-Cas9-sgGria or control bilaterally in NAc (left). Representative image of virus spread through fluorescent in situ hybridization Cas9 probe binding to viral DNA (magenta) and ChAT mRNA expression (green) (right). **(o)** Performance in the 2AA task by group. Mixed-effects group effect: p = 0.0154 (sgGria1 n = 11, empty n = 12 mice). **(p)** Avoidance probability across trials with best fit model probability by group. **(q)** Best-fit parameters by group with bootstrap estimated error. Bootstrap permutation tests for group difference: Learning p = 0.680; Value p = 0.024 (sgGria1 n = 11, empty n = 12 mice). Data are shown as mean ± s.e.m. *p < 0.05, **p < 0.01, ns: p > 0.05. Schematics created in BioRender. Macdonald, E. (2026) https://BioRender.com/4pm04zs. Source data are provided as a Source Data file. All statistical tests are two-sided.

while the contralateral hemisphere received a control virus lacking the sgRNA (Cas9-empty) (Fig. 5A). Mice were then trained in the 2AA task, and acute brain slices were prepared two hours after the final training session for whole-cell recordings from CINs in each hemisphere. In the control (Cas9-empty) hemisphere, CINs displayed properties consistent with synaptic potentiation as previously observed: elevated AMPA oEPSCs, increased AMPA/NMDA ratios, greater rectification, and enhanced sensitivity to NASPM (Fig. 5D-K). In contrast, CINs in the sgGria1-injected hemisphere exhibited significantly reduced AMPA-mediated currents, lower AMPA/NMDA ratios, and decreased rectification indices (Fig. 5D-K). NASPM sensitivity was also attenuated, consistent with a failure to incorporate GluA2-lacking AMPA receptors in the absence of GluA1 (Fig. 5L,M). Importantly, these differences were not observed in untrained home-cage controls (Supplementary Fig. 8), indicating that GluA1 deletion does not alter baseline synaptic strength but specifically prevents training-induced plasticity. Together, these results demonstrate that GluA1 expression in CINs is required for the synaptic potentiation at PVT → CIN synapses induced by safety learning.

We next asked how this plasticity contributes to behavior. In principle, synaptic strengthening at PVT → CIN synapses could support either the acquisition of avoidance actions or the assignment of motivational value to successful avoidance outcomes. To dissociate these possibilities, we bilaterally injected ChAT-Cre mice with the *Gria1*-targeting CRISPR construct and trained them in the 2AA task. Compared to control animals, deletion of GluA1 in CINs significantly

impaired avoidance performance (Fig. 5N,O; Supplementary Fig. 9). To determine whether this deficit reflected disrupted action learning or altered outcome valuation, we applied RL models to trial-by-trial behavior. Strikingly, GluA1 deletion selectively reduced the estimated value parameter associated with successful avoidance, while leaving the learning rate of avoidance actions intact (Fig. 5P,Q). This pattern closely mirrored the effects of PVT→NAc silencing, directly implicating CIN plasticity as a critical node for encoding safety-related value. Individual-level fits using Bayesian sampling further attributed the group difference to altered initial values rather than differences in locomotion or general performance, as confirmed by open-field testing (Supplementary Fig. 9).

Together, these findings identify GluA1-dependent synaptic plasticity at PVT → CIN synapses as a key mechanism through which avoidance enhances the motivational value of safety. By strengthening the ability of thalamic input to recruit cholinergic modulation and downstream dopaminergic signaling, this local plasticity enables internal goal representations, here the value of successful avoidance, to gain stable influence over striatal motivational circuitry and sustain safety-seeking behavior.

## Discussion

Many adaptive behaviors depend on the ability to maintain and prioritize internal goals whose motivational significance extends beyond the immediate presence of sensory cues[54]. These goals may arise from internal state, prior experience, or the outcomes of action,

and must be stabilized over time to sustain motivated behavior. Here, we identify a thalamostriatal circuit mechanism through which the PVT supports this process. Using safety-seeking behavior as a tractable example, our findings show how PVT engagement of striatal cholinergic and dopaminergic signaling stabilizes the motivational value of an internal goal and promotes behavioral persistence. This extends classical models of valuation, which emphasize external cues and innate reinforcers[28,55] by demonstrating how thalamic circuits enable internally generated goals to exert sustained control over action.

Our findings support a conceptualization of the PVT as part of a central system for stabilizing and prioritizing internal goal representations and broadcasting value-related signals to guide behavior. Depending on downstream target, this broadcast can serve distinct computational functions. For instance, PVT projections to the amygdala assign motivational significance to sensory events, shaping cue–outcome associations[56]. Within the NAc, PVT outputs can likewise engage distinct interneuron populations to modulate behavioral policy: PVT input to parvalbumin interneurons has been shown to impose a threat- and drug-sensitive brake on reward seeking[57,58], whereas the PVT→NAc CIN pathway described here appears to support the maintenance and reinforcement of goal value across action and time. In this way, PVT signaling can either constrain or sustain behavior depending on internal state and circuit context. This dual capacity, to assign value to external cues and to elevate the value of internally generated outcomes, positions the PVT as a key node translating internal demands into motivational control. As such, our results support and extend the emerging view of the thalamus not only as an outward-facing "searchlight" prioritizing sensory information for perception[6,59–61], but also as an inward-facing system that elevates internal goals for sustained motivational guidance[22].

PVT's role in internal goal prioritization is dynamically shaped by physiological and emotional states. Prior studies have shown that PVT engagement enhances the attractiveness of food odors during hunger[62] and suppresses reward-seeking under threat[25,57,63], highlighting its function as a flexible gate that adjusts motivational salience according to internal context[64–66]. Our findings build on this framework by identifying a synaptic mechanism through which PVT signaling stabilizes the motivational value of an internal, action-contingent outcome. In this view, safety is not treated as a special class of inferred outcome, but rather as a goal whose value must be actively maintained to sustain avoidance behavior over time and effort.

At the circuit level, we uncover a disynaptic architecture in which thalamic input recruits dopaminergic signaling through modulation of striatal CINs. CINs are well known to regulate DA release[40–42,67,68] and contribute to motivated behavior[69–71]. Recent studies further show that CINs are required for effortful reward pursuit[72] and that glutamatergic plasticity onto CINs shapes reinforcement-driven decision-making[73]. Our findings extend this framework by showing that experience-dependent strengthening of PVT→CIN synapses enhances CIN recruitment and amplifies dopaminergic output specifically in response to internally generated goals. GluA1-dependent incorporation of calcium-permeable AMPARs at these synapses may thus act as a local gain mechanism, enabling internal goal representations to exert sustained influence over striatal motivational circuitry. Interestingly, PVT synapses onto parvalbumin interneurons and Drd2-expressing medium spiny neurons in the NAc have also been reported to undergo opioid-dependent plasticity at sites enriched for calcium-permeable AMPARs[25,57,58]. Together, these observations suggest that PVT→NAc pathways may share a common plasticity motif, through which state-dependent thalamic input drives long-term modifications in striatal circuit function and behavioral policy.

Importantly, our findings should be interpreted within the broader and actively evolving literature on striatal DA–ACh interactions. While classical and contemporary studies have demonstrated that CIN activity can powerfully gate DA release through nicotinic mechanisms, particularly at moments of action and outcome[40–45,69], other work has found little cholinergic regulation of DA signals in vivo, especially during cue-driven or low-effort behaviors[74–76]. Recent work by Touponse and colleagues[71] provides an important framework for reconciling these views, demonstrating that cholinergic modulation of DA is strongly context-dependent and preferentially engaged when behavioral demands require sustained motivation, internal state integration, or effortful control. In this framework, CIN–DA coupling is not a default property of striatal processing, but a conditional mechanism recruited when internal goals must be actively stabilized to guide behavior. Our data align closely with this emerging perspective. In the shuttle avoidance task, cholinergic and dopaminergic signals are initially weakly coupled but become increasingly coordinated as animals learn to sustain avoidance based on internally represented safety value. The temporal ordering we observe, where CIN activation reliably precedes DA release during the safety period, supports a model in which cholinergic signaling serves as an enabling or amplifying step for dopaminergic engagement, rather than a redundant or parallel signal. Thus, rather than contradicting studies reporting independent DA and ACh dynamics, our findings suggest that PVT-driven plasticity selectively recruits CIN-mediated dopaminergic modulation under conditions where internal goals, rather than external cues, dominate behavioral control.

Our computational analyses further refine this interpretation. Reinforcement learning models identified a selective impact on outcome value: across optogenetic perturbations of PVT→NAc signaling, both our RW-based fits and Bayesian inference converged on reduced safety value ($\beta$) as the dominant explanation for impaired avoidance persistence, with comparatively preserved action learning parameters. In contrast, for the Gria1 deletion in CINs, Bayesian modeling yielded a more nuanced pattern: $\beta$ showed a non-significant downward trend, whereas the most reliable group difference emerged as a reduction in V1 (initial value). Importantly, these outcomes are not mutually exclusive. Rather, they suggest that PVT → CIN plasticity can influence safety-seeking through at least two computational entry points: by modulating online valuation in the moment (as revealed by acute optogenetic perturbations) and by shaping the baseline motivational set-point or initial value assigned to avoidance (most evident in the Gria1 manipulation). This interpretation is consistent with the broader idea that thalamostriatal mechanisms help stabilize internal goal value while allowing downstream learning systems to update behavior trial-by-trial[2].

Finally, these circuit mechanisms may have broad relevance for psychiatric disease. Disruption of thalamostriatal plasticity could distort the prioritization of internal goals, either inflating their motivational weight, as in pathological avoidance, or attenuating it, as observed in the amotivation and avolition associated with schizophrenia[77–80]. Postmortem studies have revealed reduced expression of cholinergic interneuron markers in the striatum of individuals with schizophrenia, suggesting a link between CIN dysfunction and impaired motivational control[81], and recent work in a genetic mouse models has established a mechanistic link between impaired thalamostriatal transmission, CIN dysfunction, and avolition-like phenotypes[78]. By linking synaptic plasticity to the stabilization of internal goal value, our findings offer a mechanistic framework for understanding how motivational priorities are constructed, and how they become dysregulated, in psychiatric disease.

Importantly, while our experiments focused on avoidance-derived safety, the circuit logic revealed here is unlikely to be specific to defensive behavior. The PVT has been implicated across a broad range of motivated states, including reward seeking, drug relapse, feeding, and motivational vigor[21,25,57,58,82]. In this broader view, PVT→NAc signaling may serve as a general mechanism for stabilizing internally defined goals, whether aversive or appetitive, by assigning them sustained motivational weight and supporting their continued

control over action selection and vigor. Rather than encoding the specific content of a goal, the PVT may contribute a higher-order function: preserving the relevance of goal representations once they have been established through experience, internal state, or prior outcomes. Through plastic interactions with striatal cholinergic and dopaminergic systems, this thalamostriatal pathway could therefore support persistence, effort, and internal prioritization across motivational domains.

Together, our findings identify a thalamostriatal plasticity mechanism through which internal goals acquire durable motivational weight. By strengthening PVT → CIN transmission and recruiting cholinergic-dependent dopaminergic output in the NAc, this circuit provides a route for learned goals to exert stable motivational control when external incentives are weak or absent.

## Methods

### Subjects

All procedures were performed in accordance with the Guide for the Care and Use of Laboratory Animals and were approved by the National Institute of Mental Health (NIMH) Animal Care and Use Committee. Mice used in this study were group housed under a 12-h light/dark cycle (6:00–18:00 light), at temperatures of 70–74 °F and 40–65% humidity, with food and water available *ad libitum*. After surgery, mice were singly housed. C57BL/6NJ strain mice (The Jackson Laboratory #005304) were crossed with Drd2-Cre (GENSAT founder line ER44), ChAT-Cre (The Jackson Laboratory # 018957), or DAT-Cre (#006660). Experiments were performed in adult male and female mice (8-20 weeks). Sex was considered in the study design; however, the study was not powered a priori to detect sex-specific effects. Therefore, data were pooled across sex for all analyses. Animals were randomly allocated to the different experimental conditions reported in this study.

### Viral vectors

AAV9-hSyn-Flex-GCaMP8s-WPRE (#162377), pAAV.Syn.-Flex.GCaMP6s.WPRE.SV40 (#100845), pAAV-hSyn-dLight1.2 (#111068), pAAV-hsyn-GRAB_rDA1m (#140556), pAAV-hSyn-DIO-mCherry (#50459-AAV2/9), pAAV-hSyn-hChR2(H134R)-EYFP (#26973-AAV9), pAAV-Syn-ChrimsonR-tdT (#59171-AAV9), pAAV-hSyn-DIO-hM4D(Gi)-mCherry (#44362-AAV2), and pAAV-EF1a-DIO-ChrimsonR-mRuby2-KV2.1-WPRE-SV40 (#124603) were purchased from Addgene. pAAV-FLEX-SaCas9-U6-sgGria1 (#124853), pAAV-FLEX-SaCas9-U6-sgRNA (empty) (#124844) were purchased as plasmids from Addgene and packaged into AAV2/9 at Vector Biolabs. AAV2-Ef1α-DIO-eNpHR3.0-mCherry (Deisseroth) and AAV2-syn-FLEX-ChrimsonR-tdTomato (Boyden) were produced by the Vector Core of the University of North Carolina. AAV9-hsyn-gACh4h(Ach3.8) was purchased from WZ Biosciences. All viral vectors were stored in aliquots at −80 °C until use.

### Stereotaxic surgery

All stereotaxic surgeries were conducted as described in our animal study protocol using previously described procedures[24] and stereotaxic coordinates[27]. Mice were first anesthetized with a ketamine (100 mg/kg) plus xylazine (10 mg/kg) solution and placed in a stereotaxic device (AngleTwo, Leica Biosystems). The following stereotaxic coordinates were targeted for viral injections and/or optical fiber implantation: NAc, 1.65 mm from bregma, ±.9 mm lateral from the midline, and -4.6 mm vertical from the cortical surface, 0° angle for photometry, 8° angle for optogenetics. PVT, −1.60 mm from bregma, 0.06 mm lateral from midline and −3.30 mm vertical from cortical surface, 6.12° angle for both fiber photometry and optogenetics. VTA, -3.08 mm from bregma, ±.5 mm lateral from the midline, and -4.51 mm vertical from the cortical surface. For fiber photometry and optogenetic experiments, an optical fiber (400 μm for photometry, Doric

Lenses/Inper; 200 μm for optogenetics, Thorlabs/Inper) was implanted over the target immediately after viral injections and cemented using Metabond Cement System (Parkell) and Jet Brand dental acrylic (Lang Dental Manufacturing). For head-fixed training, a head bar was attached with light-cure cement (3 M RelyX) on top of Metabond. After all surgical procedures, animals were returned to their home cages and placed on a heating pad for 3 hours for post-surgical recovery and monitored for 72 h with diet supplements. Animals received subcutaneous injections with Metacam (meloxicam, 5 mg/kg) for analgesia and anti-inflammatory purposes. Mice without correct targeting of optical fibers or vectors were excluded from this study.

### Behavior

**2-way active avoidance.** Mice were trained on the 2AA task, replicating previous methodological procedures[27,83] and are detailed below. The behavioral apparatus consisted of a custom-built shuttle box (18 cm × 36 cm × 30 cm) that contained two identical chambers separated by a hurdle (17.5 cm × 6 cm). The hurdle projected 3 cm above the floor and allowed mice easy access to both chambers. The floor consisted of electrifiable metal rods (H10-11M-TC-SF, Coulbourn Instruments) and was connected to a shock generator (H13-15, Coulbourn Instruments). Before each subject was trained/tested, the shuttle box was wiped clean with 70% ethanol. The mouse's behavior was captured with a USB camera during each session. A speaker located on the top of the shuttle box (50 cm high) was used to deliver the WS. Subjects' movement and TTLs of WS, US and optogenetic stimulation were recorded and coordinated by ANY-maze versions 5-7 (Stoelting).

After a 5-min habituation period, mice were trained with daily sessions of 2AA, each consisting of 30 presentations of the WS (4 kHz, 75 dB, lasting up to 15 s each). Trials in which subjects failed to shuttle to the adjacent chamber before the termination of the WS resulted in the presentation of the unconditioned stimulus (US, 0.6 mA foot shock lasting up to 7 s each) until subjects escaped to the opposite chamber (escape trials). No subject failed to escape the US. For trials in which subjects shuttled to the opposite chamber during the WS, the WS was abruptly terminated, and the US was also prevented (avoidance trials). The inter-trial interval (ITI) was random between 30 and 40 s for all experiments except Supplementary Fig. 4, where the ITI was increased to 60 s. The avoidance rate was calculated as the percentage of the number of avoidance trials over the total number of trials. For optogenetic and fiber photometry experiments fiber patch cords were attached every session of training. For all photometry experiments, subjects that did not reach 30% avoidance rates by Day 3 were excluded from data analysis. All experiments were conducted in awake, freely moving mice in test chambers during the light cycle.

**Extinction with response prevention.** As shown in Supplementary Fig. 5, after five days of 2AA training, subjects underwent one session of extinction with response prevention. During this session, animals were confined to a single chamber of the 2AA apparatus using a custom transparent plexiglass divider and presented with 15 WS (15 s each 35-45 s intertrial interval; ITI). On subsequent extinction days, the divider was removed, allowing animals to freely shuttle between chambers while receiving 30 WS presentations (15 s, 30–40 s ITI). No shocks were delivered during extinction, and shuttling did not terminate the WS. To verify that extinction-related changes in photometry signals reflected neural activity rather than photobleaching, a reinstatement session was conducted using the standard 2AA protocol.

**Probabilistic avoidance with surprise safety omissions.** As shown in Supplementary Fig. 1, following five days of standard 2AA training, mice underwent a single test session designed to introduce rare violations of learned safety contingencies. Animals were counterbalanced to pre-defined probabilistic and deterministic shuttling directions.

Each session began with a 20-trial baseline phase identical to training. After this baseline, animals entered a probabilistic phase in which successful shuttling continued to terminate the WS in both directions; however, in the preassigned probabilistic direction, a subset of successful avoidances was followed by delivery of a shock at WS offset (surprise safety omissions). In contrast, the preassigned deterministic direction remained fully reliable, such that successful avoidance was never followed by shock.

Surprise safety omission trials were restricted to the probabilistic direction and were pseudorandomly distributed with constraints designed to prevent consecutive surprise trials (implemented as five surprise events separated by 5–10 intervening trials). This schedule resulted in an average of $62 \pm 2$ total trials per session, an average gap of eight trials between surprise shocks, and an effective shock probability of $22.4 \pm 5\%$ for successful avoidance in the probabilistic direction.

All task parameters were otherwise identical to those used during standard 2AA training.

**Head fixed avoidance.** The behavioral apparatus comprised a 20 cm diameter foam cylinder connected to a rotary encoder, a head-fixation frame, and a holder for a compressed air nozzle positioned behind the animal, directed toward the base of the tail. The rotary encoder was linked to an interface board connected to the Bpod system (Sanworks, NY). Air puff delivery was controlled through a picospritzer triggered by Bpod TTL output, with pressure set to 50 psi.

Mice were trained daily for 3 days, each session comprising 50 trials. On each trial, a WS (12 kHz, 75 dB, up to 10 s) was presented. If the subject failed to run at least 5 cm on the wheel before WS termination, the US (air puff, up to 5 s) was delivered. If the running requirement was met during the WS, the WS was immediately terminated and the US omitted (avoidance trials). The ITI was 30 s. The avoidance rate was calculated as the percentage of avoidance trials over the total number of trials. Animals typically reached achieved stable performance ($\geq 60\%$ avoidance) by training day 3 or 4.

For the progressive ratio (PR) test, the maximum WS duration was increased to 20 s. The required distance for avoidance started at 25 cm and increased in 5 cm increments. Advancement to the next stage required two consecutive successful avoidance trials. The session ended after six consecutive US deliveries, and the breakpoint was defined as the longest distance at which avoidance was successfully achieved.

## Fiber photometry

Fiber photometry was performed in accordance with previously described methodological procedures[84] and are detailed below. Mice were allowed to habituate to the fiber patch cord in their home cage for approximately 5 min before each behavior test. GCaMP fluorescence and isosbestic autofluorescence signals were excited by the fiber photometry system (Doric Lenses) using two sinusoidally modulated LEDs (473 nm at 211 Hz and 405 nm at 311 Hz) controlled by a standalone driver (DC4100, ThorLabs). Both LEDs were combined via a commercial Mini Cube fiber photometry apparatus (Doric Lenses) into a fiber patch cord (400-µm core, 0.48 NA) connected to the brain implant in each mouse. The light intensity at the interface between the fiber tip and the animal was adjusted from 10 µW to 20 µW (but was constant throughout each test session for each mouse). An RZ5P fiber photometry acquisition system with Synapse v98 software (Tucker-Davis Technologies) collected and saved real-time demodulated emission signals and behavior-relevant TTL inputs. For each trial, signals (F473 nm) were compared with autofluorescence signals (F405 nm) to control for movement and bleaching artifacts. Signal data were de-trended by first applying a least-squares linear fit to produce $F_{\text{fitted 405 nm}}$, and dF/F was calculated as $(F_{473 \text{ nm}} - F_{\text{fitted 405 nm}})/F_{\text{fitted 405 nm}}$. All signal data are presented as the z-score of the dF/F from baseline

(pre-WS/pre-Event) segments. In Supplementary Fig. 7, a third channel was added (560 nm LED driven at 711 Hz). This signal was compared to autofluorescence signals (F405 nm) as described. To visualize event-aligned signals, trial-based events such as cue onset and safety were identified from TTL cue and shock inputs to the RZ5P amplifier from Any-Maze, and events such as shuttle initiation were aligned to video-tracking output using custom R code. Mean signal is the average across subject averages Z-score normalized to a 5 s pre-event baseline. Area under the curve was calculated with trapezoid approximation to baseline (pre) and event (post) time periods.

For Fig. 3 and Supplementary Fig. 7, a RZ10x fiber photometry acquisition system (Tucker-Davis Technologies) with integrated drivers, LEDs (405 nm, 465 nm, 560 nm) and photosensors was used to collect and demodulate emission signals with analogous protocols and analysis. All photometry recordings utilized a custom Doric mini cube for wavelength separation. Sensors including GCaMP, dLight1.2, and gACh3.8 were recorded using 473 nm and 465 nm drivers filtered through a 460-490 nm input channel and were paired with outputs from a 500–540 nm channel. A short-wavelength control channel used 405 nm drivers filtered through 400–410 nm input and demodulated with 420-450 nm output to sensors. 560 nm drivers for the rDA1m sensor utilized the 550–580 nm input channel, and signals were demodulated from outputs through the 600-680 nm channel. The 600-680 nm channel served as an input for optogenetic stimulation using red light.

**Fiber photometry with optogenetic and chemogenetic manipulations.** In Fig. 2, mice with dLight in NAc unilaterally and halorhodopsin or mCherry in PVT underwent 2AA training. Fiber photometry recordings of dLight were taken during training on days 1, 3, and 5. In addition, a red light (Opto Engine 635 nm) was triggered at the completion of all avoidance trials concurrent with WS offset for a 5 s duration.

In Fig. 3K–O, mice were injected with dLight in NAc, DREADDs (Gi) in NAc ChAT+ neurons, and ChrimsonR or mCherry in PVT. Fiber photometry recordings of dLight were taken in a novel context on separate days during optogenetic stimulation of PVT terminals. For within-mouse comparisons, each subject was given a saline injection I.P. 30 min prior to recording. The following day the same subjects were given a CNO injection (5 mg/kg) I.P. 30 minutes prior to recording. The mice were trained in 2AA for 5 days before repeating saline and CNO recording conditions.

## Optogenetics

In Fig. 1 and Supplementary Fig. 4, mice were subjected to six 2AA sessions (one session per day), on training days 1-5 optogenetic stimulation was triggered at the completion of all avoidance trials concurrent with WS offset for a 5 s duration. The light was powered on for 5 s for experiments with halorhodopsin stimulation, or at 20 Hz, 20% duty cycle for 5 s in ChrimsonR experiments, and light intensity at the interface between the fiber tip and the implant cannula was ~10 mW. Bilateral optogenetic manipulations used a yellow (Ce:YAG + LED Driver 565 nm, Doric) or green light (Opto Engine 561 nm), while optogenetics multiplexed with fiber photometry used a red light (Opto Engine 635 nm) as described above to limit off-target excitation.

## Electrophysiology

Mice were trained in 2AA for 5 days, and two hours following the last session, the subjects were then sacrificed for slice recordings by an experimenter blind to conditions. For yoked controls, each was paired with a trained subject. The pairs were trained concurrently wherein the yoked subject received the same duration of both WS and shock.

Acute brain slices were prepared according to previously described protocols[24,83,85]. Mice were anesthetized with isoflurane and transcardially perfused with an ice-cold NMDG cutting solution

containing (in mM): 92 mM N-Methyl-D-glucamine, 2.5 mM KCl, 1.25 mM NaH2PO4, 10 mM MgSO4, 0.5 mM CaCl2, 30 mM NaHCO3, 20 mM glucose, 20 mM HEPES, 2 mM thiourea, 5 mM Na-ascorbate, 3 mM Na-pyruvate, at 7.3-7.4 pH, saturated by 95% O2 and 5% CO2. Coronal sections (300 μm thickness) containing the NAc were cut in the ice-cold NMDG cutting solution using a VT1200S automated vibrating-blade microtome (Leica Biosystems), and were subsequently transferred to a heated incubation chamber containing the NMDG cutting solution at 34-35 °C. After recovery for 12 min, slices were transferred to a room temperature (20-24 °C) holding chamber containing a HEPES-modified artificial cerebrospinal fluid (92 mM NaCl, 2.5 mM KCl, 1.25 mM NaH2PO4, 2 mM MgSO4, 2 mM CaCl2, 30 mM NaHCO3, 25 mM glucose, 20 mM HEPES, 2 mM thiourea, 5 mM Na-ascorbate, 3 mM Na-pyruvate, at 7.3 pH, saturated by 95% O2 and 5% CO2) and remained in the holding chamber until recordings.

Whole-cell patch-clamp recordings were conducted under visual guidance using an Olympus BX51 microscope with transmitted light illumination, and ChAT+ neurons were identified based on their fluorescence (mCherry). Recordings from ChAT+ neurons in the NAc were obtained with Multiclamp 700B amplifiers (Molecular Devices), and data were filtered at 1-2 kHz using a 1440 A Digidata Digitizer and Clampex software (Molecular Devices). For recordings, slices were transferred to the recording chamber and constantly supplied with a room-temperature ACSF (118 mM NaCl, 2.5 mM KCl, 26.2 mM NaHCO3, 1 mM NaH2PO4, 20 mM glucose, 2 mM MgCl2, and 2 mM CaCl2, at pH 7.4, saturated by 95% O2 and 5% CO2, flow rate = 2.0-2.5 ml/min). Pharmacological agents were added to the ACSF and bath applied. All recordings were made with borosilicate glass pipettes with tip resistance of 3-6 MΩ. pipettes were filled with Cs-based internal solution containing (in mM): 117 mM Cs methanesulfonate, 10 mM HEPES, 2.5 mM MgCl2, 2 mM Na2-ATP, 0.4 mM Na2-GTP, 10 mM Na2-phosphocreatine, 0.6 mM EGTA, 5 mM QX-314 at pH 7.2 and 288-290 mOSM. Optogenetically evoked synaptic responses were achieved by illuminating the acute slices with blue LED (470 nm, CoolLED) drive ChR2-expressing terminals. Light stimulation was delivered every 15 seconds and synaptic responses were recorded at holding potentials of −60 mV for AMPA-receptor-mediated responses and +40 mV for NMDA-receptor-mediated responses. NMDA-receptor-mediated responses were quantified as the EPSC magnitude at +40 mV (50 ms following stimulation). Evoked EPSCs were recorded with picrotoxin (100 μM) added to the ACSF. To assess presynaptic function, a paired-pulse stimulation protocol (50 ms inter-stimulus interval) was used to evoke double-EPSCs, and the paired-pulse ratio (PPR) was quantified as the ratio of the peak amplitude of the second EPSC to that of the first EPSC. Rise time and decay time of EPSCs were measured between 20–80% of amplitude, respectively. For the measurement of rectification of AMPA current, spermine (100 μM) was added in the internal solution, while picrotoxin (100 μM) and NMDA receptor blocker D-(-)-2-Amino-5-phosphonopentanoic acid (AP-5) (50 μM) were added in ACSF. Neurons were held at −70, −40, 0, +40 or +70 mV, respectively. The rectification index (RI) was calculated by dividing the absolute amplitude of average EPSC at +40 mV by that at −70 mV. Naspm trihydrochloride (NASPM, 100 μM, Tocris) was applied to the slice while stimulating every 15 s. A stable baseline was recorded for at least 5 min followed by a 10 min application of NASPM. The average amplitude of the EPSC was calculated during the 5 min period prior to NASPM application (baseline) and the average of the four stimulations starting 9 min after drug application, with the EPSC amplitude normalized to the baseline period.

## Fast-scan cyclic voltammetry
Fast-scan cyclic voltammetry was performed as previously described[32]. Carbon-fiber electrodes (CFEs) were prepared with a cylindrical carbon-fiber (7 mm diameter, ~150 mm of exposed fiber) inserted into a glass pipette and were conditioned with an 8-ms-long triangular voltage ramp (-0.4 to 1.2 and back to -.4 V vs Ag/AgCl reference at 400 V/s) delivered every 15 ms. CFEs showing current >1.8 mA or <1.0 mA in response to the voltage ramp at ~0.6 V were discarded. During recording, CFEs were held at -0.4 V versus Ag/AgCl, and a triangular voltage ramp was delivered every 100 ms. DA signals were evoked by single pulse optical stimulations, where a fiberoptic (200 mm diameter, 0.22 NA, ThorLabs) connected to a blue LED (470 nm, 1.8 mW, ThorLabs) was placed over the carbon fiber and light pulses (~0.6 ms) were delivered every 2 min. Data were collected with a retrofit headstage (CB-7B/ EC with 5 MX resistor) using a Multiclamp 700B amplifier (Molecular Devices) after a low-pass filter at 3 kHz and digitized at 100 kHz using a DA board (NI USB-6229 BNC, National Instruments). Data acquisition and analysis were performed using a custom-written software, VIGOR, in Igor Pro (Wavemetrics) using mafPC (courtesy of MA Xu-Friedman). The current peak amplitudes of the evoked DA transients were converted to DA concentration according to the post-experimental calibration using 1-3 mM DA.

## Histology and immunofluorescence
Histological procedures were conducted as previously described[84,86]. Briefly, to confirm viral expression and optic fiber placement, animals were anesthetized with euthanasia solution (Vet One) and underwent transcardial perfusion with PBS (pH 7.4, 4 °C), followed by paraformaldehyde (PFA) solution (4% in PBS, 4 °C). Extracted brains were immersion-fixed in 4% PFA (4 °C, ≥2 h), then transferred to a 30% sucrose in PBS solution for cryoprotection until fully equilibrated (24−36 h). A Leica freezing microtome (SM 2010R) was used to collect 50 μm coronal sections. Sections were stained with DAPI, cover slipped with laboratory-prepared mounting medium and imaged using a Nikon AR1 confocal.

**RNAscope in situ hybridization.** RNAscope was performed as previously described[86,87]. For Fig. 5 and Supplementary Fig. 7, brains from adult male C57BL/6NJ mice (8-12 weeks) were rapidly fresh-frozen and cut into 16 μm coronal sections on a cryostat (Leica Biosystems), mounted onto Superfrost Plus slides (Fisher Scientific), and stored at −80 °C. Sections covering the posterior extent of the PVT were fixed in 4% paraformaldehyde (30 min at 4 °C), dehydrated through graded ethanol solutions (50%, 70%, and 100% ×2), and RNAscope™ Multiplex Fluorescent v2 reagents (Advanced Cell Diagnostics) were applied as per the manufacturer's guidelines. Following hydrogen peroxide treatment (10 min) and Protease IV incubation (30 min at room temperature), sections were hybridized with probes targeting Chat (#408731) and saCas9 (#501621) (2 h at 40 °C). Signal amplification (Amp 1–4) and HRP-based detection were performed at 40 °C, followed by fluorophore development using Opal 520 and Opal 570 reagents. Following DAPI counterstaining, sections were coverslipped using laboratory-prepared mounting medium and imaged using a Zeiss Axioscan slide-scanning microscope.

## Statistical analysis and reproducibility
All data were plotted and analyzed with GraphPad Prism (v10.4.0), Python (v3.11.4), MATLAB (R2023b), RStudio (v2023.09.1), or Clampfit 10 (Molecular Devices). Behavioral tracking during the 2-way signaled active was performed using TopScan software (CleverSys), as previously described[27,83]. Change-point analyses were performed using the R package *changepoint* (URL:https://CRAN.R-project.org/package=changepoint). R code used for active avoidance behavioral and photometry analysis is publicly available at: https://github.com/Penzolab/Data-analysis-of-Two-way-active-avoidance-task.git. (DOI: 10.5281/zenodo.12707790). A complete summary table of statistical results is provided (Supplementary Data 1). All statistical tests were two-sided and are described in the corresponding figure legends and Supplementary Data 1. Data distribution were assessed for normality using Shapiro-Wilk tests. For datasets that violated assumptions of normality, non-parametric approaches or mixed modeling were used as appropriate. Unbalanced area-under-the-curve (AUC) datasets were

analyzed using linear mixed-effects models fitted with restricted maximum likelihood (REML) using statsmodels mixedlm in Python. Models included a random intercept for subject to account for repeated measures. Main effects of categorical factors were assessed using Wald F-tests on model-estimated contrasts. Pairwise comparisons were performed using t-tests on contrasts and p-values adjusted with the Sidak method for multiple comparisons. Balanced datasets were tested with mixed-effects models (REML) in GraphPad Prism. Sample sizes are typically the same or exceed those estimated by power analyses (power = 0.80, $\alpha = 0.05$). For fiber photometry experiments alone, the sample size is 5-10 mice. For fiber photometry experiments with optogenetic and/or chemogenetic manipulations, the sample size is 4-6 mice. For optogenetic behavior experiments, the sample size is 7-13 mice. For ex vivo electrophysiology experiments, the sample size was 5-22 cells from 3-5 mice. Independent replication confirmed consistent results across experiments, except for experiments described in Figs. 2g–k and 3d–o, where the desired sample sized was obtained in one cohort. Randomized assignment was used for all experiments. For ex vivo patch-clamp electrophysiology recordings, investigators were blinded to allocation during experiments. All 2AA photometric signals and behavioral performance were analyzed blinded.

**RL model.** A Rescorla-Wagner reinforcement learning model[88] was fit to the sample mean using Matlab:

$$p_a(t+1) = p_a(t) + \alpha\left(\beta - p_a(t)\right) \tag{1}$$

where $p_a$ is the probability of avoidance, t is the trial, $\alpha$ is the learning rate, and $\beta$ is the set value of avoidance. $\alpha$, $\beta$, and a starting $p_a$ were fit as free parameters using least sum of squares. Standard error was estimated within groups using a 1000 bootstrap permutation test with replacement. A 5000 bootstrap permutation test with a null distribution generated by randomly reassigning group labels was used to assess the significance of differences between group parameters and p values determined by two-tailed comparison to the group parameter differences. The simulation in Fig. 1K was created using extreme high and low $\alpha$ and $\beta$ parameters seen in real datasets. Trial outcome was generated using softmax probability (inverse temperature = 0.8) and 100 simulations were averaged.

**RL model fit to individuals.** A Bayesian hierarchical model (CmdStanPy (version 1.3.0[89]) was used to fit individual subject behavior with a Rescorla-Wagner learning model. 20000 sampling iterations were run on four chains and 10000 warm-up samples were discarded. Convergence diagnostic R-hat values were satisfactory for all parameters in each model fit. A graphical representation of the Bayesian hierarchical model is depicted in Supplementary Fig. 3A with the prior distributions. Parameter recovery was performed using parameters from subject fits (Supplementary Fig. 3B).

**Schematics**
Cartoon schematics were created in BioRender and licensed under: Macdonald, E. (2026) https://BioRender.com/4pm04zs. Histological maps were adapted from Allen Institute Mouse Brain Reference Atlas[90].

**Reporting summary**
Further information on research design is available in the Nature Portfolio Reporting Summary linked to this article.

## Data availability
The datasets generated in this study have been deposited in the Zenodo database under accession code DIO:10.5281/zenodo.20031828. All relevant processed data are also provided in the Supplementary Information/Source Data file. The datasets used in this study are fully publicly accessible without restriction. Source data are provided with this paper.

## Code availability
R code used to analyze active avoidance behavior, and photometric signal is available at the following repository: https://github.com/Penzolab/Data-analysis-of-Two-way-active-avoidance-task.git. https://doi.org/10.5281/zenodo.12707790.

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

## Acknowledgements
The authors thank the members of the Section on the Neural Circuits of Emotion and Motivation and the Laboratory on Neural Circuits and Behavior of NIMH who offered scientific feedback. In addition, we thank the NIMH IRP Rodent Behavioral Core and the NIMH IRP Systems Neuroscience Imaging Resource for supporting this work. Finally, we thank Dr. Michael Halassa for offering scientific and writing feedback.

## Author contributions
Conceptualization: E.E.M. and M.A.P.; Data acquisition: E.E.M., J.M., D.L., K.Y., M.E.A., R.A.W., Ya.L., and H.C.G.; Data analysis: E.E.M., D.L., K.Y., and M.E.A.; Provided the GRAB ACh3.8 sensor: G.L. and Yu.L. Project management and supervision: M.A.P., B.B.A., and V.A.A.; Writing original draft: E.E.M and M.A.P.

## Funding
This work was supported by the Intramural Research Program (IRP) of the National Institutes of Health (NIH): NIMH IRP 1ZIAMH002950 (to M.A.P.) and ZIA MH002928 (to B.B.A). The contributions of the NIH author(s) were made as part of their official duties as NIH federal employees, are in compliance with agency policy requirements, and are considered Works of the United States Government. However, the findings and conclusions presented in this paper are those of the author(s) and do not necessarily reflect the views of the NIH or the U.S. Department of Health and Human Services.

## Competing interests
The authors declare no competing interests.
