## [Transparent Peer Review file · Nature Communications]

A synaptic mechanism for encoding the learned value of action-derived safety

Corresponding Author: Dr Mario Penzo

Version 0:

Reviewer comments:

Reviewer #1

(Remarks to the Author)

This is a conceptually and technically novel and impressive study, wherein the authors examine the role of a PVT-NAc circuit in learning safety. They use and combine optogenetics, photometry, slice physiology, and computational modeling to demonstrate that PVT to NAc cholinergic interneuron (CIN) synapses are potentiated via Glu2A-lacking AMPA receptor insertion, and this plasticity is thought to mediate the learning of value of safety to drive avoidance.

I really enjoyed the paper. It is elegant, novel, and likely to be very impactful for the field. However, I have some concerns that I believe are critical for the authors to address, related to the RL model interpretations and power, integrating the study into a more complex field regarding PVT-NAc function, and a missing control in the Gria1 knockdown experiment.

Major

1. The authors claim manipulations of the PVT-NAc CIN circuit specifically impair the learning of safety's value. There seem to be two issues with this conclusion that are critical to address:

- a. First, the data is far from convincing. For example, in Figure 1L, the claim is made based on a barely significant effect for "R" (safety value; $p = 0.0448$) and no effect of "L" (action learning), though it is nearly trending ($p = 0.1032$) and has a substantial amount of variance. The same is true for Figure 5Q, which suffers from insufficient animals per group and even more variance. Furthermore, the channelrhodopsin data in Extended Data Figure 2 suggest that L is indeed influenced by this circuit (even though this is a facilitation experiment, it lends credence to the point that the R vs. L dissociation in Figure 1 and 5 lacks power and conclusions that value, but not learning, are mediated by the PVT-NAc CIN circuit may be premature). On a related note, authors should present individual data points in the bar graphs when analyzing R and L, akin to the other bar graphs of the manuscript, such that the reader can fully appreciate the influence of variance on the RL results.
- b. Second, the experiments don't seem to be usable for the RL model since the confound of expression was not accounted for. The effects observed could simply be due to a loss of expression, induction of aversion, etc., that may have little to do with the safety value (R). A key experiment, which I suggest they perform for Figure 1 experiments, is providing a no-manipulation probe test to determine if the behavior comes back (and thus is an effect on expression) or not (and thus is an effect on learning as predicted).

2. A significant oversight of the paper is its failure to acknowledge a large body of work showing that the PVT-NAc pathway can do the complete opposite, stop behavior rather than promote an action. Other labs, including but not limited to work from the Otis and Do-Monte labs, have shown this effect which seems to be a fundamentally opposing to the current findings. The distinctions are likely related to different PVT neurons or downstream neurons (e.g., CIN vs PV interneurons) that are being recruited, which has been proposed to enable 'go' vs 'stop' signals (e.g., McGinty & Otis 2020 FIBN). In the current state, the paper provides an oversimplified picture and misses key opportunities to synthesize the literature into a more complete model of how the PVT provides control over behavior.

3. The paper's evidence that Gria1 KO in CINs prevents learning-induced synaptic adaptations is missing the key control of learning itself. The authors compare trained mice with the gene knockout to trained mice without. It's possible that deleting Gria1 simply impairs the synapse from the start. If that's the case, the "lack of plasticity" isn't a failure to adapt, but rather pre-

existing due to Gria1 KO. This ambiguity further weakens their conclusion that they have identified the specific mechanism for learning value.

Minor

1. It is challenging to see differences in several of the violin plots, due to the size of the figures, spread of the data, and colors of the median bar which is the same as the violin itself. In some cases I can't tell from the graphs alone if even the significant effects are up or down.

2. Figure 5Q is not in the figure caption and is merged with Figure 5P.

Reviewer #2

(Remarks to the Author)

This study aimed to dissect a thalamostriatal circuit for the processing of active avoidance, or “safety”, learning and performance. The authors combine a series of state-of-the-art techniques and demonstrate that activity in PVT neurons that project to the NAc medial shell increases at times of active-avoidance. This increase in PVT-NAc activity drives behavioral performance as well as acetylcholine and cholinergic-dependent dopamine release, with active avoidance training leading to excitatory plasticity in PVT-NAc/CIN synapses mediated by calcium-permeable AMPA receptors.

This is an interesting paper that is very clearly structured and written. It provides valuable new information on the mechanisms of active harm avoidance/ safety seeking, and offers a promising demonstration of the behavioral relevance of a nicotinic receptor-dependent local dopamine release mechanism in the striatum. However, the interpretation of the data in the context of the used behavioral task is highly irregular and needs to be addressed.

The major issue is that it is unclear to this reviewer why “safety”, as defined within the confines of the performed experiments, would be an “internally constructed”, “inferred”, or “abstract” goal when it is clearly defined by the changes in sensory states set by the warning signal (no shock when WS terminates = safety). For the presented task to isolate a truly inferred process, either the delivery of foot shock would need to be probabilistic (so that the mouse would have to infer their actions resulted in safety despite non-contingent shock omissions) or the shock would have to be entirely unsigned. While it is still interesting that the dissected circuit seems to contribute to punishment-avoidance learning in general, the argument that this study uncovered “a circuit mechanism by which such internally constructed outcomes acquire motivational value” is not supported by the presented data. Indeed, congruent with the idea that the type of “safety” studied here is just absence of punishment, the modelling procedures used here are just a simple Rescorla-Wagner model, where “safety value” R is the standard US maximal conditioning values.

On the modelling, it is overly simplistic and lacks rigor. For starters, the authors only seem to have modelled the safety learning component of the task, despite the design involving both a cue-shock association learning as well as a response-safety component. Moreover, it is not clear why the fit was applied to the sample mean instead of the individual animal data. This could also be obscuring individual variation in the contribution of the PVT-NAc pathway to task behavior. This is especially important given the extremely high error estimates from their bootstrapping procedures in the optogenetic inhibition groups. Not having individual fits also preclude the authors from applying standard parameter recovery procedures, which are important for the rigorous implementation of data fits.

Finally, while there has been extensive *in vitro* evidence on nicotinic receptor-mediated dopamine release in the striatum, there is also controversy as to its relevance in behavior. The authors do not discuss recent work that suggest that *in vivo* dopamine and acetylcholine dynamics in the striatum are largely independent (e.g., Krok et al, *Nature*, 2023; Chantranupong et al, *Nature*, 2023; Costa et al, *Current Biology*, 2025). While the authors did not simultaneously record dopamine and acetylcholine in their experiments (which would have been a strength, but is not strictly necessary for their conclusions), the manuscript would be substantially strengthened if they show a more detailed analysis comparing the time course of cholinergic and dopaminergic signals during the behavioral epochs in their task. For example, this could indicate whether it is likely that acetylcholine is directly driving dopamine responses, or perhaps enhances responses that already initiated. As a related but minor point, in the dLight data, the WS onset induces a sharp reduction in dopamine signal that precedes the shuttle action. It is not clear how the authors correct for this when performing baseline correction and estimating the subsequent dopamine responses.

Reviewer #3

(Remarks to the Author)

Version 1:

Reviewer comments:

Reviewer #1

(Remarks to the Author)

The authors have done a great job responding to my prior critiques. I only have minor edits and suggestions.

Minor Edits:

Figure 1C-E) Figure legend for Fig. 1C to E are out of order with figure presented.

Figure 1F-G) Figure legend should indicate that the left of the plot for AUC quantification is individual values and right is the mean and SEM for each day. This applies to all other cases where AUC quantification is shown.

Suggestions:

Figure 1D and EDF6B) Indicate the number of trials on the Y-axis

EDF1B) Error bars on test day are not visible, consider setting axis from 0.4-1 to more clearly plot data.

Methods: Add the wavelength used to record Dlight.

Methods: Add the wavelength used to stimulate ChrimsonR

Figure 4: Did all CIN neurons respond to PVT terminal stim, and did this differ between groups? Consider adding this information.

Reviewer #2

(Remarks to the Author)

The authors have done a very good job of responding to my concerns. I have nothing critical to add.

Reviewer #4

(Remarks to the Author)

DEPARTMENT OF HEALTH & HUMAN SERVICES
Intramural Research Program
National Institute of Mental Health
Bethesda Maryland 20814

Mario A. Penzo PhD.
Telephone (Office): 301-451-7296
mario.penzo@nih.gov

Dear Reviewers,

Thank you for your thoughtful, detailed, and constructive evaluation of our manuscript. We were particularly encouraged by your recognition of the conceptual novelty, technical rigor, and potential impact of this work. Your critiques were extremely helpful in identifying areas where the framing, analyses, and experimental controls could be strengthened. In response, we have substantially revised the manuscript, performed additional experiments, and expanded our analytical approaches. Collectively, these revisions have significantly improved the clarity and rigor of the study and further strengthen support for our original conclusions.

Below, we first provide a summary of the major revisions and new experiments added in response to your comments. This is followed by a point-by-point response in which each critique is reproduced in full and addressed individually (in **bold**).

Sincerely,

Mario A. Penzo, Ph.D.
Senior Investigator
Section on the Neural Circuits of Emotion and Motivation
National Institute of Mental Health Intramural Research Program
National Institutes of Health

Summary of major revisions and new experiments:

In response to the reviewers' comments, we have substantially revised the manuscript and conducted additional experiments and analyses that directly address the conceptual, analytical, and mechanistic concerns raised. Major changes include:

1. **Comprehensive revision of the Introduction and Discussion.** We completely rewrote the Introduction to clarify the conceptual framework of the study, shifting emphasis from “inference” toward internal valuation and goal representation, and to better situate our findings within the broader literature on thalamic control of motivation. We also expanded and revised the Discussion to provide a more scholarly synthesis of prior work on

PVT→NAc circuits, including studies demonstrating both facilitatory and suppressive roles of this pathway, as well as recent literature on NAc neurotransmitter dynamics.

2. **Addition of a Day 6 light-off probe test for optogenetic silencing of PVT→NAc input (Figure 1).** To distinguish effects on behavioral expression from effects on learning or outcome valuation, we included a no-manipulation probe session following outcome-locked optogenetic inhibition. This experiment demonstrates that avoidance deficits persist in the absence of stimulation, supporting an effect on motivational valuation rather than transient expression.
3. **Implementation of Bayesian modeling and individual-subject fits across experiments.** We extended our computational analyses to include Bayesian approaches and reinforcement-learning fits at the level of individual subjects, rather than relying solely on group means. These analyses were applied consistently across optogenetic and genetic experiments, allowing explicit assessment of inter-subject variability and parameter recovery.
4. **Inclusion of a probabilistic avoidance task.** To directly address concerns regarding reliance on warning-signal offset as a sensory readout of safety, we introduced a probabilistic version of the task in which successful avoidance no longer guaranteed shock omission. This experiment demonstrates sensitivity to outcome uncertainty and supports the interpretation that avoidance behavior reflects valuation of action-dependent outcomes rather than simple sensory feedback.
5. **Increased sample size for the CIN-specific CRISPR/Cas9 behavioral experiment (Figure 5).** We expanded the cohort size for the Gria1 conditional knockout experiment, improving statistical power for behavioral and modeling analyses and strengthening conclusions regarding the role of CIN plasticity in safety valuation.
6. **Addition of a baseline synaptic control experiment for sgGria1-injected mice.** To rule out pre-existing synaptic deficits, we performed new slice electrophysiology experiments assessing baseline AMPA/NMDA currents in naïve sgGria1-injected mice. These data demonstrate that Gria1 deletion does not alter baseline synaptic strength, but selectively prevents learning-induced plasticity.
7. **Dual in vivo recordings of acetylcholine and dopamine.** We performed simultaneous ACh and DA recordings in the NAc to directly assess their temporal relationship during avoidance behavior. These experiments reveal a training-dependent temporal coupling in safety signal peak activity.

Reviewer comments and responses:

Reviewer #1 (Remarks to the Author):

This is a conceptually and technically novel and impressive study, wherein the authors examine the role of a PVT-NAc circuit in learning safety. They use and combine optogenetics, photometry, slice physiology, and computational modeling to demonstrate that PVT to NAc cholinergic interneuron (CIN) synapses are potentiated via Glu2A-lacking AMPA receptor insertion, and this plasticity is thought to mediate the learning of value of safety to drive avoidance.

I really enjoyed the paper. It is elegant, novel, and likely to be very impactful for the field. However, I have some concerns that I believe are critical for the authors to address, related to the RL model interpretations and power, integrating the study into a more complex field regarding PVT-NAc function, and a missing control in the Gria1 knockdown experiment.

We thank the reviewer for recognizing the novelty and impact of this study. We have addressed the reviewers concerns by adding a supporting model to reinforce the interpretation of our previous, added additional animals to the Gria1 knockdown behavior experiment, and performed the suggested control experiment for the Gria1 electrophysiology experiment.

1. The authors claim manipulations of the PVT-NAc CIN circuit specifically impair the learning of safety's value. There seem to be two issues with this conclusion that are critical to address:

a. First, the data is far from convincing. For example, in Figure 1L, the claim is made based on a barely significant effect for “R” (safety value; $p = 0.0448$) and no effect of “L” (action learning), though it is nearly trending ($p = 0.1032$) and has a substantial amount of variance. The same is true for Figure 5Q, which suffers from insufficient animals per group and even more variance. Furthermore, the channelrhodopsin data in Extended Data Figure 2 suggest that L is indeed influenced by this circuit (even though this is a facilitation experiment, it lends credence to the point that the R vs. L dissociation in Figure 1 and 5 lacks power and conclusions that value, but not learning, are mediated by the PVT-NAc CIN circuit may be premature). On a related note, authors should present individual data points in the bar graphs when analyzing R and L, akin to the other bar graphs of the manuscript, such that the reader can fully appreciate the influence of variance on the RL results.

We thank the reviewer for this careful and fair assessment of the modeling results. We agree that the original presentation did not sufficiently convey variance or statistical power, and that stronger support was needed before advancing conclusions regarding differential effects on value (R – renamed to β) versus action learning (L – renamed to α).

To directly address these concerns, we made several substantive changes. First, we increased the cohort size for the CIN-specific Gria1 CRISPR/Cas9 behavioral experiment (**Figure 5**), substantially improving statistical power. Given finite time for revisions, we chose to repeat the experiment in **Figure 5N-Q** over **Figure 1H-L** because we have the additional experiment to support that PVT-NAc inhibition reduces safety outcome value in **Figure 1M-O**. Second, we now perform reinforcement-learning fits at the level of individual subjects, rather than fitting only to group-averaged data. This allowed us to explicitly assess inter-subject variability, better evaluate the robustness of the observed effects, and display individual data points for all β and α parameters, enabling direct visualization of variance as suggested (**Extended Data Figures 3, 4, & 9**). Both models were included in support of each other, where the simpler fit to means was left in the main figures. These fit parameters do not have individual points to display, as the error is estimated through bootstrapping.

Across both optogenetic and genetic manipulations, these revised analyses reveal a consistent pattern in which perturbation of the PVT-NAc-CIN pathway preferentially reduces the estimated value of successful avoidance, while effects on action learning are comparatively modest. Importantly, we have tempered our language throughout the manuscript to avoid claims of strict exclusivity, and instead emphasize preferential involvement in outcome valuation under the conditions tested.

b. Second, the experiments don't seem to be usable for the RL model since the confound of expression was not accounted for. The effects observed could simply be due to a loss of expression, induction of aversion, etc., that may have little to do with the safety value (R). A key experiment, which I suggest they perform for **Figure 1** experiments, is providing a no-manipulation probe test to determine if the behavior comes back (and thus is an effect on expression) or not (and thus is an effect on learning as predicted).

We agree with the reviewer that distinguishing effects on behavioral expression from effects on learning or valuation is critical for interpreting the modeling results. To directly address this concern, we performed the suggested no-manipulation probe test.

Specifically, we added a Day 6 light-off session following outcome-locked optogenetic inhibition of PVT-NAc input during training. As shown in the revised **Figure 1J**, avoidance deficits persisted during this probe session despite the absence of optogenetic manipulation. This persistence indicates that the manipulation produced a lasting alteration in the motivational value assigned to successful avoidance, rather than a transient disruption of performance, aversion induction, or loss of behavioral expression.

This new experiment has been incorporated into the Results, and its implications for avoidance valuation are discussed (**Lines 105-108**).

2. A significant oversight of the paper is its failure to acknowledge a large body of work showing that the PVT-NAc pathway can do the complete opposite, stop behavior rather than promote an action. Other labs, including but not limited to work from the Otis and Do-Monte labs, have shown this effect which seems to be a fundamentally opposing to the current findings. The distinctions are likely related to different PVT neurons or downstream neurons (e.g., CIN vs PV interneurons) that are being recruited, which has been proposed to enable 'go' vs 'stop' signals (e.g., McGinty & Otis 2020 FIBN). In the current state, the paper provides an oversimplified picture and misses key opportunities to synthesize the literature into a more complete model of how the PVT provides control over behavior.

We thank the reviewer for highlighting this important literature. In the revised Discussion, we now explicitly integrate prior work demonstrating both facilitatory and suppressive roles of PVT-NAc signaling, including studies from the Otis and Do-Monte laboratories. We frame our findings within a broader model in which the PVT broadcasts motivationally relevant information to the NAc, with downstream cell-type-specific mechanisms (e.g., CINs versus other interneurons) enabling bidirectional control over behavior (Lines 306-313).

3. The paper's evidence that Gria1 KO in CINs prevents learning-induced synaptic adaptations is missing the key control of learning itself. The authors compare trained mice with the gene knockout to trained mice without. It's possible that deleting Gria1 simply impairs the synapse from the start. If that's the case, the "lack of plasticity" isn't a failure to adapt, but rather pre-existing due to Gria1 KO. This ambiguity further weakens their conclusion that they have identified the specific mechanism for learning value.

We agree entirely with this concern and thank the reviewer for raising it. We performed the requested control experiment assessing baseline PVT-CIN synaptic strength in naïve sgGria1-injected and control mice. As shown in new Extended Data Figure 8, baseline AMPA/NMDA ratios were unchanged, demonstrating that Gria1 deletion does not impair synaptic strength at baseline but selectively prevents learning-induced plasticity.

Minor Comments:

1. It is challenging to see differences in several of the violin plots, due to the size of the figures, spread of the data, and colors of the median bar which is the same as the violin itself. In some cases I can't tell from the graphs alone if even the significant effects are up or down.

We thank the reviewer for pointing this out. We agree that, in some cases, the violin plots did not optimally convey directionality or variance, particularly at the reduced scale required for multi-panel figures.

To address this concern, we revised the relevant figures to display individual data points beside bar plots (mean \pm s.e.m.), allowing clear visualization of both effect direction and

variability across subjects or cells. This representation preserves summary statistics while making it easier to assess the distribution and magnitude of the effects directly from the figures. We believe this change substantially improves figure clarity and addresses the reviewer's concern.

2. Figure 5Q is not in the figure caption and is merged with Figure 5P.

This has been corrected.

Reviewer #2 (Remarks to the Author):

This study aimed to dissect a thalamostriatal circuit for the processing of active avoidance, or “safety”, learning and performance. The authors combine a series of state-of-the-art techniques and demonstrate that activity in PVT neurons that project to the NAc medial shell increases at times of active-avoidance. This increase in PVT-NAc activity drives behavioral performance as well as acetylcholine and cholinergic-dependent dopamine release, with active avoidance training leading to excitatory plasticity in PVT-NAc/CIN synapses mediated by calcium-permeable AMPA receptors.

This is an interesting paper that is very clearly structured and written. It provides valuable new information on the mechanisms of active harm avoidance/ safety seeking, and offers a promising demonstration of the behavioral relevance of a nicotinic receptor-dependent local dopamine release mechanism in the striatum. However, the interpretation of the data in the context of the used behavioral task is highly irregular and needs to be addressed.

We thank the reviewer for recognizing the clarity, technical rigor, and behavioral relevance of the study. We also appreciate the careful critique of our conceptual framing and modeling approach, which prompted substantial revisions to both the manuscript and the experimental design. In response, we revised the conceptual emphasis of the paper, added new behavioral, computational, and physiological experiments, and expanded the analytical framework. These changes not only address the reviewer's concerns but also strengthen and reaffirm the central conclusions of the study.

1. The major issue is that it is unclear to this reviewer why “safety”, as defined within the confines of the performed experiments, would be an “internally constructed”, “inferred”, or “abstract” goal when it is clearly defined by the changes in sensory states set by the warning signal (no shock when WS terminates = safety). For the presented task to isolate a truly inferred process, either the delivery of foot shock would need to be probabilistic (so that the mouse would have to infer their actions resulted in safety despite non-contingent shock omissions) or the shock would have to be entirely unsigned.

We agree with the reviewer that our original framing overemphasized “inference” and did not sufficiently distinguish between sensory correlates of the task and the motivational process under investigation. In response, we substantially revised the Introduction and Discussion to shift the conceptual emphasis from inference to internal valuation and goal representation.

In the revised manuscript, we no longer argue that safety in this task is abstract or inferred in the absence of sensory structure. Instead, we emphasize that safety is an internally valued, action-contingent outcome whose motivational significance must be learned and maintained across time and effort.

Importantly, we also performed the reviewer’s suggested probabilistic avoidance experiment, in which successful shuttling no longer guarantees shock omission. Under these conditions, animals still perform avoidance, but with altered latencies, demonstrating sensitivity to outcome uncertainty. This finding argues against a strategy based solely on warning-signal termination and supports the interpretation that avoidance behavior reflects valuation of an action-dependent outcome rather than simple sensory feedback. Critically, while warning-signal offset is a salient sensory event, behavioral performance in the task depends on the reliability and predictability of the action–outcome contingency rather than on any single sensory feature, as demonstrated in Extended Data Figure 1.

Together, these conceptual and experimental revisions directly address the reviewer’s concern and clarify that our focus is on how internally valued goals, here, action-derived safety, acquire and retain motivational significance.

2. While it is still interesting that the dissected circuit seems to contribute to punishment-avoidance learning in general, the argument that this study uncovered “a circuit mechanism by which such internally constructed outcomes acquire motivational value” is not supported by the presented data. Indeed, congruent with the idea that the type of “safety” studied here is just absence of punishment, the modelling procedures used here are just a simple Rescorla-Wagner model, where “safety value” R is the standard US maximal conditioning values.

On the modelling, it is overly simplistic and lacks rigor. For starters, the authors only seem to have modelled the safety learning component of the task, despite the design involving both a cue-chock association learning as well as a response-safety component. Moreover, it is not clear why the fit was applied to the sample mean instead of the individual animal data. This could also be obscuring individual variation in the contribution of the PVT-NAc pathway to task behavior. This is especially important given the extremely high error estimates from their bootstrapping procedures in the optogenetic inhibition groups. Not having individual fits also preclude the authors from applying standard parameter recovery procedures, which are important for the rigorous implementation of data fits.

We appreciate this critique and agree that the original modeling approach required both greater rigor and clearer justification. In response, we substantially expanded the computational analyses.

First, we retained the original fits to the sample mean, as these provide an intuitive visualization of the learning dynamics at the group level. However, to directly address concerns regarding individual variability and parameter stability, we additionally implemented a Bayesian hierarchical version of the same Rescorla–Wagner model. Importantly, this approach preserves the same update rule with analogous parameters, but estimates subject-level parameters from group-level prior distributions. The Bayesian approach provides stable parameter estimates even for poor avoiders while preserving individual variability. However, fitting to individuals has its own caveats to consider, where poor avoiders drove the learning parameter to near zero, making both α and β more difficult to interpret for those subjects. Therefore, we included both models in support of each other.

Crucially, the hierarchical Bayesian fits yielded results comparable to those obtained with the original mean-based fits. Across both optogenetic and genetic perturbations of the PVT→NAc pathway, we consistently observe preferential reductions in safety-related value parameters, with comparatively preserved action learning rates. This convergence across analytic approaches indicates that the observed group differences are not artifacts of fitting instability or averaging procedures.

To further ensure robustness, we performed parameter recovery analyses using simulated datasets generated from the fitted parameters (including poor avoiders). These analyses confirm reliable parameter identifiability and demonstrate that the reported group differences are not driven by estimation noise (Extended Data Figures 3, 4, and 9).

Second, we clarified the scope of the model. The model is not intended to capture all components of avoidance learning (e.g., cue–shock acquisition), but rather to dissociate action learning from outcome valuation once avoidance behavior is established. Across both optogenetic and genetic perturbations of the PVT→NAc pathway, the hierarchical model converges on preferential reductions in safety-related value parameters, with comparatively preserved action learning rates. This pattern is consistent across Figures 1 and 5 and is further supported by effort-based breakpoint data (Fig. 1N–O).

We have revised the manuscript to clarify these points and temper language where appropriate, while maintaining the core conclusion that PVT→NAc signaling preferentially supports outcome valuation under the conditions tested.

3. Finally, while there has been extensive in vitro evidence on nicotinic receptor-mediated dopamine release in the striatum, there is also controversy as to its relevance in behavior. The

authors do not discuss recent work that suggest that in vivo dopamine and acetylcholine dynamics in the striatum are largely independent (e.g., Krok et al, Nature, 2023; Chantranupong et al, Nature, 2023; Costa et al, Current Biology, 2025). While the authors did not simultaneously record dopamine and acetylcholine in their experiments (which would have been a strength, but is not strictly necessary for their conclusions), the manuscript would be substantially strengthened if they show a more detailed analysis comparing the time course of cholinergic and dopaminergic signals during the behavioral epochs in their task. For example, this could indicate whether it is likely that acetylcholine is directly driving dopamine responses, or perhaps enhances responses that already initiated.

We agree with the reviewer that the relationship between acetylcholine and dopamine dynamics in vivo remains an area of active debate, and that establishing behavioral relevance requires careful temporal and circuit-level evidence. In response to this concern, we performed within-subject, simultaneous recordings of acetylcholine and dopamine in the nucleus accumbens during avoidance behavior.

These new experiments reveal three key findings. First, during the safety period, cholinergic transients reliably precede dopaminergic peaks, and this temporal offset becomes more pronounced with avoidance training (Extended Data Fig. 7). Second, this temporal ordering is consistent with our prior results showing that PVT→NAc input is required for safety-related dopamine release (Fig. 2) and that PVT-evoked dopamine release depends on intact cholinergic signaling (Fig. 3). Third, chemogenetic silencing of CINs abolishes PVT-evoked dopamine following training, demonstrating that CIN recruitment is not merely correlative but necessary for translating thalamic input into dopaminergic output.

Importantly, we do not interpret these data as evidence that acetylcholine obligatorily drives dopamine release. Instead, consistent with recent studies highlighting partial independence between DA and ACh dynamics in vivo e.g. (Krok et al., 2023; Chantranupong et al., 2023; Costa et al., 2025), we propose that cholinergic signaling functions as a permissive or amplifying gate that enables dopaminergic responses under conditions requiring sustained motivation and internal goal representation.

In support of this interpretation, a recent study by (Touponse et al., 2026) demonstrated that CIN activity in the nucleus accumbens regulates effort-related and motivational aspects of dopamine signaling during instrumental behavior. Notably, while that study did not directly test thalamic inputs, the authors speculated that long-range excitatory afferents, including thalamic projections, may provide a key source of motivational drive to CINs. Our data directly extend this framework by identifying the PVT as a specific upstream circuit that engages CINs through learning-dependent synaptic plasticity to regulate dopamine release during safety valuation.

We have incorporated these new data and an expanded discussion of DA–ACh interactions into the Results and Discussion (Lines 340-358).

Minor Comments:

1. As a related but minor point, in the dLight data, the WS onset induces a sharp reduction in dopamine signal that precedes the shuttle action. It is not clear how the authors correct for this when performing baseline correction and estimating the subsequent dopamine responses.

We thank the reviewer for raising this important methodological point. In the analyses presented in the manuscript, photometry signals were baseline-normalized to the period immediately preceding each aligned event, as now clarified in the Methods (Lines 770-773). This approach ensures that safety-related dopamine responses are estimated relative to the local signal state at the time of outcome onset.

To explicitly rule out the possibility that the safety-period dopamine signal reflects a rebound from cue-evoked suppression rather than a genuine increase in release, we performed additional analyses in which dLight signals were z-score normalized to the inter-trial interval (ITI) preceding cue onset. Under this normalization, dopamine signals during the safety period remain significantly elevated above zero, demonstrating that the safety-related signal represents a genuine transient increase in dopamine release rather than a return to baseline following cue-induced decreases. These effects are also visible in the trial-by-trial heatmaps in Extended Data Figure 6.

We have incorporated this clarification into the Methods and Results. Consistent with our findings, increases in dopamine release during safety or punishment-avoidance periods have also been reported by other groups using complementary techniques such as fast-scan cyclic voltammetry and microdialysis (Oleson et al., 2012; Stelly et al., 2019; Dombrowski et al., 2013) in addition to dopamine sensors (Lopez et al., 2025).

Reviewer #3 (Remarks to the Author):

We thank the reviewer for taking the time to evaluate this study.

DEPARTMENT OF HEALTH & HUMAN SERVICES

Intramural Research Program
National Institute of Mental Health
Bethesda Maryland 20814

Mario A. Penzo PhD.
Telephone (Office): 301-451-7296
mario.penzo@nih.gov

REFERENCES:

- Chantranupong, L., Beron, C. C., Zimmer, J. A., Wen, M. J., Wang, W., & Sabatini, B. L. (2023). Dopamine and glutamate regulate striatal acetylcholine in decision-making. *Nature*, *621*(7979), 577–585. <https://doi.org/10.1038/s41586-023-06492-9>
- Costa, K. M., Zhang, Z., Deutsch, D., Zhuo, Y., Li, G., Li, Y., & Schoenbaum, G. (2025). Dopamine and acetylcholine correlations in the nucleus accumbens depend on behavioral task states. *Current Biology*, *35*(6), 1400-1407.e3. <https://doi.org/10.1016/j.cub.2025.01.064>
- Dombrowski, P. A., Maia, T. V., Boschen, S. L., Bortolanza, M., Wendler, E., Schwarting, R. K. W., Brandão, M. L., Winn, P., Blaha, C. D., & Da Cunha, C. (2013). Evidence that conditioned avoidance responses are reinforced by positive prediction errors signaled by tonic striatal dopamine. *Behavioural Brain Research*, *241*, 112–119. <https://doi.org/10.1016/j.bbr.2012.06.031>
- Krok, A. C., Maltese, M., Mistry, P., Miao, X., Li, Y., & Tritsch, N. X. (2023). Intrinsic dopamine and acetylcholine dynamics in the striatum of mice. *Nature*, *621*(7979), 543–549. <https://doi.org/10.1038/s41586-023-05995-9>
- Lopez, G. C., Van Camp, L. D., Kovaleski, R. F., Schaid, M. D., Sherathiya, V. N., Cox, J. M., & Lerner, T. N. (2025). Region-specific nucleus accumbens dopamine signals encode distinct aspects of avoidance learning. *Current Biology*, *35*(10), 2433-2443.e5. <https://doi.org/10.1016/j.cub.2025.04.006>
- Oleson, E. B., Gentry, R. N., Chioma, V. C., & Cheer, J. F. (2012). Subsecond Dopamine Release in the Nucleus Accumbens Predicts Conditioned Punishment and Its Successful Avoidance. *Journal of Neuroscience*, *32*(42), Article 42. <https://doi.org/10.1523/JNEUROSCI.3087-12.2012>
- Stelly, C. E., Haug, G. C., Fonzi, K. M., Garcia, M. A., Tritley, S. C., Magnon, A. P., Ramos, M. A. P., & Wanat, M. J. (2019). Pattern of dopamine signaling during aversive events predicts active avoidance learning. *Proceedings of the National Academy of Sciences*, *116*(27), Article 27. <https://doi.org/10.1073/pnas.1904249116>
- Touponse, G. C., Pomrenze, M. B., Yassine, T., Denomme, N., Wang, M., Mehta, V., Zhang, Z., Malenka, R. C., & Eshel, N. (2026). Cholinergic modulation of dopamine release drives effortful behaviour. *Nature*. <https://doi.org/10.1038/s41586-025-10046-6>

DEPARTMENT OF HEALTH & HUMAN SERVICES
Intramural Research Program
National Institute of Mental Health
Bethesda Maryland 20814

Mario A. Penzo PhD.
Telephone (Office): 301-451-7296
mario.penzo@nih.gov

Dear Reviewers,

Thank you for your continued time, effort, and careful evaluation of our revised manuscript. We greatly appreciate the thoughtful and constructive feedback provided across both rounds of review. Your earlier comments were instrumental in strengthening the clarity, rigor, and overall presentation of the work, and we believe the manuscript has been significantly improved as a result. We are encouraged that the current assessment does not raise any major concerns and are grateful for the additional minor suggestions, particularly regarding figure presentation. We have carefully addressed all of these points and implemented the recommended revisions and appreciate the reviewers' thorough re-evaluation and helpful guidance in refining the manuscript further.

Below, we first provide a summary of the minor revisions made in response to your comments. This is followed by a point-by-point response in which each critique is reproduced in full and addressed individually (in **bold**).

Sincerely,

Mario A. Penzo, Ph.D.
Senior Investigator
Section on the Neural Circuits of Emotion and Motivation
National Institute of Mental Health Intramural Research Program
National Institutes of Health

Summary of major revisions and new experiments:

In response to the reviewers' comments, we have revised the manuscript to address the concerns raised. Changes include figure edits and methods additions.

REVIEWERS' COMMENTS

Reviewer #1 (Remarks to the Author):

The authors have done a great job responding to my prior critiques. I only have minor edits and suggestions.

We thank Reviewer #2 for their time and careful re-evaluation of the revised manuscript. We have taken steps to address your minor edits and suggestions.

Minor Edits:

Figure 1C-E) Figure legend for Fig. 1C to E are out of order with figure presented.

Thank you for catching this mistake. It has been corrected.

Figure 1F-G) Figure legend should indicate that the left of the plot for AUC quantification is individual values and right is the mean and SEM for each day. This applies to all other cases where AUC quantification is shown.

This has been clarified in figure legends.

Suggestions:

Figure 1D and EDF6B) Indicate the number of trials on the Y-axis

Since the heatmaps of trial signals are in order of avoidance latency and not trial order, we thought that including trial numbers might confuse the readers. Instead, we have now included the number of trials for each plot in the legend for Figures 1D, 3G, and Supp. Data Fig. 6B (formerly EDF6B).

EDF1B) Error bars on test day are not visible, consider setting axis from 0.4-1 to more clearly plot data.

Thank you for this suggestion. We noticed for Extended Data Figure 1B that our y-axis labels were incorrect (avoidance probability and not percent) and this was corrected. We attempted to change the axis and size of the plot, but the error in the points without visible error is so small (the largest among them being 1.2%) the axis rescaling did not correct this issue. We left the plot in its previous version and the source data is provided.

Methods: Add the wavelength used to record Dlight.

Methods: Add the wavelength used to stimulate ChrimsonR

We have clarified our methods section and now include a description for the wavelengths used for photometry recordings in lines 686-697. The description of optogenetic design was reworded to improve clarity and include stimulation wavelengths in lines 712-719.

Figure 4: Did all CIN neurons respond to PVT terminal stim, and did this differ between groups?

DEPARTMENT OF HEALTH & HUMAN SERVICES
Intramural Research Program
National Institute of Mental Health
Bethesda Maryland 20814

Mario A. Penzo PhD.
Telephone (Office): 301-451-7296
mario.penzo@nih.gov

Consider adding this information.

Thank you for raising this important point. In Figure 4, we report CIN postsynaptic responses to optogenetic stimulation of PVT terminals recorded from 22 neurons across 4 trained mice and 19 neurons across 3 yoked control mice. No cells were excluded due to lack of response. While oEPSC amplitudes varied (as shown in Figure 4E), all recorded neurons exhibited measurable responses, with the smallest amplitude being 24.1 pA in a cell from a control animal.

Importantly, recordings were obtained from CINs in the NAc shell in regions where fluorescently labeled PVT terminals were clearly visible within the same area. Variability in postsynaptic current amplitude may reflect differences in the local density of ChR2-expressing PVT terminals, but as the experimenter was blind to training condition, it is unlikely that any group differences arose from bias in cell selection. Overall, our data suggest that most, if not all, NAc CINs in proximity to PVT terminals receive functional PVT input. However, we cannot exclude the possibility that a subpopulation of CINs lacking such input exists outside the sampled regions.

Reviewer #2 (Remarks to the Author):

The authors have done a very good job of responding to my concerns. I have nothing critical to add.

We are grateful to Reviewer #2 for their assessment of the revised manuscript.

Reviewer #4 (Remarks to the Author):

We are grateful to Reviewer #3 for their assessment of the revised manuscript.